# GUARANTEED APPROXIMATION BOUNDS FOR MIXED-PRECISION NEURAL OPERATORS

**Renbo Tu**[*1], **Colin White**[*2], **Jean Kossaifi**[3], **Boris Bonev**[3], **Gennady Pekhimenko**[1],
**Kamyar Azizzadenesheli**[3], **Anima Anandkumar**[2]

[1] University of Toronto, [2] Caltech, [3] NVIDIA

## ABSTRACT

Neural operators, such as Fourier Neural Operators (FNO), form a principled approach for learning solution operators for partial differential equations (PDE) and other mappings between function spaces. However, many real-world problems require high-resolution training data, and the training time and limited GPU memory pose big barriers. One solution is to train neural operators in mixed precision to reduce the memory requirement and increase training speed. However, existing mixed-precision training techniques are designed for standard neural networks, and we find that their direct application to FNO leads to numerical overflow and poor memory efficiency. Further, at first glance, it may appear that mixed precision in FNO will lead to drastic accuracy degradation since reducing the precision of the Fourier transform yields poor results in classical numerical solvers. We show that this is not the case; in fact, we prove that reducing the precision in FNO still guarantees a good approximation bound, when done in a targeted manner. Specifically, we build on the intuition that neural operator learning inherently induces an approximation error, arising from discretizing the infinite-dimensional ground-truth input function, implying that training in full precision is not needed. We formalize this intuition by rigorously characterizing the approximation and precision errors of FNO and bounding these errors for general input functions. We prove that the precision error is asymptotically comparable to the approximation error. Based on this, we design a simple method to optimize the memory-intensive half-precision tensor contractions by greedily finding the optimal contraction order. Through extensive experiments on different state-of-the-art neural operators, datasets, and GPUs, we demonstrate that our approach reduces GPU memory usage by up to 50% and improves throughput by 58% with little or no reduction in accuracy.

## 1 INTRODUCTION

Real-world problems in science and engineering often involve solving systems of partial differential equations (PDEs) (Strang, 2007). These problems typically require a fine grid for numerical solvers to guarantee convergence, e.g., climate modeling and 3D fluid dynamics simulations, and the full-scale solutions to these real-world systems are out of reach even for the world's largest supercomputers (Schneider et al., 2017).

To overcome the computational challenges of numerical solvers, fast surrogates using machine learning have been developed. Among them, *neural operators* are a powerful *data-driven* technique for solving PDEs (Li et al., 2020a; 2021a; Kovachki et al., 2023; Lu et al., 2021). Neural operators learn maps between function spaces, and they can be used to approximate the solution operator of a given PDE. The input and output functions in neural operators can be at any resolution or on any mesh, and the output function can be evaluated at any point in the domain; therefore, neural operators are *discretization convergent*: once the neural operator is trained, it can be evaluated, without any retraining, at any resolution, and it converges to a unique limit under mesh refinement (Kovachki et al., 2023). By learning from discretized data from input and solution functions, the trained models can perform inference orders of magnitude faster than traditional PDE solvers (Kovachki et al., 2023). In particular, the Fourier Neural Operator (FNO) and its extensions have been successful in solving

---

*Equal contribution. Email to: `renbo.tu@mail.utoronto.ca`.

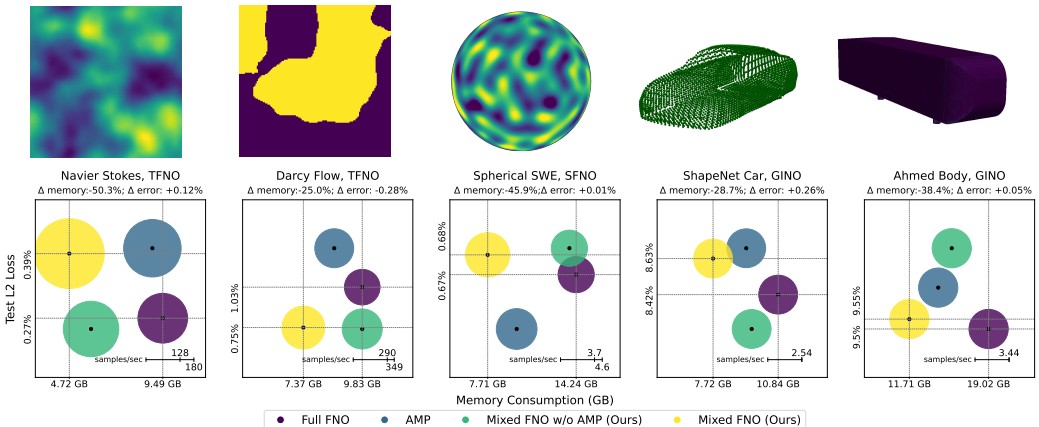

Figure 1: **Top: example data points from each dataset**: Navier-Stokes, Darcy Flow, Spherical Shallow Water, Shape-Net Car CFD, and Ahmed-body CFD. **Bottom: performance of our method compared to full-precision and AMP on each dataset.** For each dataset, we plot test error (y-axis) and GPU memory (x-axis), and we annotate the maximum throughput (proportional to the area of each ball). All data are measured on the same hardware (RTX 3090 Ti) and the same virtual environment. Memory decreases by up to 50%, while $L^2$ loss increases by at most 0.28%.

PDE-based problems with significant speedups (Li et al., 2021a; 2023; Bonev et al., 2023; Kossaifi et al., 2023; Liu et al., 2022; Gopakumar et al., 2023).

Despite their successes, training of neural operators is still compute- and memory-intensive when faced with extremely high-resolution and large-scale problems. For conventional deep learning models, there is a wealth of knowledge on automatic mixed precision (AMP) training in order to reduce memory usage and increase computational throughput (Rakka et al., 2022). We find that their direct application to neural operators results in poor memory efficiency and numerical overflow; when applied to FNO, most memory-intensive operations are in the spectral domain, which is complex-valued and not handled well by the standard AMP, as seen in Figure 1.

Conventional wisdom may suggest that reducing precision in FNO without drastic accuracy degradation is not feasible. The presence of the Fourier transform in FNO appears to be a limitation in reducing precision since it suffers from numerical instabilities on high-frequency modes under reduced precision. Hence, numerical solvers, such as pseudo-spectral solvers, do indeed require very high precision to ensure numerical stability (Trefethen, 2000), and time-stepping leads to an accumulation of round-off errors under low precision that quickly explode. However, we show both theoretically and empirically that this is not the case for FNO and other operator learning approaches.

**Our approach:** In this work, we devise the first mixed-precision training method for neural operators and also derive approximation bounds that guarantee the expressivity of mixed-precision operators. We use the intuition that the Fourier transform within neural operators is already approximated by the discrete Fourier transform, since the training dataset is a discrete approximation of the ground-truth continuous signal. Intuitively, since we already incur approximation error from discretization, there is no need to run the discrete Fourier transform in full precision. Building on this, we show that, in a single FNO layer, the round-off error due to lower precision is small. Therefore, the overall round-off error remains small since a full FNO architecture is only made up of a few layers—much smaller than the number of steps in a classical pseudo-spectral solver where errors accumulate and explode.

We make the above intuitions concrete, and we develop the following theoretical approximation bounds: we characterize the precision error and resolution error of the FNO block, proving asymptotic bounds of $n^{-2/d}$ and $\epsilon$, respectively, for mesh size $n$, dimension $d$, and dynamic range $\epsilon$. Therefore, we justify using mixed precision since its error is comparable to the discretization error already present in FNO.

Motivated by these theoretical results, we introduce a mixed-precision training method by optimizing the memory-intensive half-precision tensor contractions. We devise a simple and lightweight greedy strategy to find the optimal contraction order, which considers vectorization for each intermediate tensor. Unfortunately, naïvely training neural operators in mixed precision leads to prohibitive numerical instability due to overflows in the FNO block. To address this issue, we study numerical

stabilization techniques for neural operators, finding that `tanh` pre-activation before each FFT consistently avoids numerical instability. Further, we incorporate additional vectorization steps for complex-valued inputs, not present in standard packages designed for real-valued networks.

We carry out extensive experiments to show a significant reduction in memory usage and improved throughput without sacrificing accuracy; see Figure 1. We consider three variants of FNO: tensorized (TFNO) (Kossaifi et al., 2023), spherical (SFNO) (Bonev et al., 2023), and geometry-informed (GINO) (Li et al., 2023). Across different datasets and GPUs, our method results in up to 58% improvement in training throughput and 50% reduction in GPU memory usage with little or no reduction in accuracy.

Furthermore, we show that our method is discretization convergent via zero-shot super-resolution experiments. Finally, we propose a precision schedule training routine, which transitions from mixed to full precision during training; we show this method achieves better than the baseline full-precision accuracy. Our mixed-precision training routine is lightweight and easily added to new neural operators. We release our codebase and all materials needed to reproduce our results at https://github.com/neuraloperator/neuraloperator.

**We summarize our main contributions below:**

- We introduce the first mixed-precision training routine for neural operators, which optimizes the memory-intensive tensor contraction operations in the spectral domain, and we propose using `tanh` pre-activations to minimize numerical instability.
- We theoretically ground our work by characterizing the precision and discretization errors of the FNO block, showing that these errors are comparable, proving that, done right, mixed-precision training of neural operators leads to little or no performance degradation.
- We empirically verify the superiority of our mixed-precision training approach on three state-of-the-art neural operators, TFNO, GINO, and SFNO, across four different datasets and GPUs. Our method uses **half the memory** and **increases training throughput up to 58%** across different GPUs with little or no reduction in accuracy ($< 0.1\%$).
- We provide an efficient implementation of our approach in PyTorch, which we open-source, along with all data needed to reproduce our results.

## 2 BACKGROUND AND RELATED WORK

**Neural Operators.** Many real-world scientific and engineering problems rely on solving partial differential equations (PDEs). As such, there is a recent focus on using machine learning-based methods to solve PDEs (Gupta et al., 2021; Bhatnagar et al., 2019; Lu et al., 2019; Adler & Öktem, 2017). However, most of these methods use standard neural networks and are therefore limited to a mapping between fixed input and output grid. In other words, they operate on a fixed, regular discretization and cannot learn the continuous mapping between function spaces. *Neural operators* are a new technique that addresses this limitation by directly learning maps between function spaces (Li et al., 2021b; 2020a;b; Kovachki et al., 2023). The input and output functions to neural operators can be in any resolution or mesh, and the output function can be evaluated at any point in the domain; therefore, neural operators are *discretization convergent*: once the neural operator is trained, it can be evaluated, without any retraining, at any resolution, and it converges to a unique limit under mesh refinement (Kovachki et al., 2023).

**FNO and extensions.** The Fourier neural operator, inspired by spectral methods, is a highly successful neural operator (Li et al., 2021a; Gopakumar et al., 2023; Renn et al., 2023; Wen et al., 2022). Let $\mathcal{A} : \{a : D_{\mathcal{A}} \to \mathbb{R}^{d_{\mathcal{A}}}\}$ and $\mathcal{U} : \{u : D_{\mathcal{U}} \to \mathbb{R}^{d_{\mathcal{U}}}\}$ denote the input and output function spaces, respectively. In this work, we consider the case where $D_{\mathcal{A}} = D_{\mathcal{U}} \subset \mathbb{R}^d$ for $d \in \mathbb{N}$. Given a dataset of pairs of initial conditions and solution functions $\{a_j, u_j\}_{j=1}^{N}$, which are consistent with an operator $\mathcal{G}(a_j) = u_j$ for all $1 \leq j \leq N$, the goal is to learn a *neural operator* $\mathcal{G}_{\theta}$ that approximates $\mathcal{G}$. The primary operation in FNO is the Fourier convolution operator, $(\mathcal{K}v_t)(x) = \mathcal{F}^{-1}(R \cdot T_K(\mathcal{F}v_t))(x)$, $\forall x \in D$, where $\mathcal{F}$ and $\mathcal{F}^{-1}$ denote the Fourier transform and its inverse, $R$ denotes a learnable transformation, $T_K$ denotes a truncation operation, and $v_t$ denotes the function at the current layer of the neural operator. We use the discrete Fast Fourier Transform (FFT) and its inverse (iFFT) to implement this operator on discrete data.

Building on FNO, Kossaifi et al. (2023) introduced the Tensorized Fourier Neural Operator (**TFNO**). Building on tensor methods, which have proven very successful in deep learning Panagakis et al. (2021; 2024), TFNO computes a tensor factorization (Kolda & Bader, 2009) of the weight tensors in the spectral domain, acting as a regularizer that boosts performance while decreasing parameter count. Additionally, two FNO-based neural operators have recently been proposed to improve performance in non-regular geometries. The Spherical Fourier Neural Operator (**SFNO**) (Bonev et al., 2023) is an extension of FNO to the spherical domain, which is achieved by the use of the spherical convolution theorem (Driscoll & Healy, 1994). The Geometry-Informed Neural Operator (**GINO**) (Li et al., 2023) is a highly efficient FNO-based architecture for solving PDEs with varying, irregular geometries. The architecture consists of an FNO, along with graph neural operators to transform the irregular grid inputs into and from regular latent grids on which FNO can be efficiently applied (Li et al., 2020a).

**Mixed-Precision Training.** Mixed-precision training of neural networks consists of reducing runtime and memory usage by representing input tensors and weights (and performing operations) at lower-than-standard precision. For example, PyTorch has a built-in mixed-precision mode called automatic mixed precision (AMP), which places all operations at `float16` rather than `float32`, with the exception of reduction operations, weight updates, normalization operations, and specialized operations (Paszke et al., 2019). `bfloat16` and `TF32` are common mixed-precision methods, yet they do not support discrete Fourier transforms, which are essential to FNOs (Kalamkar et al., 2019; Dean et al., 2012). There is a variety of work targeted at quantization for inference, yet these works do not reduce memory during training (Goel et al., 2020).

While well-studied for standard neural nets, mixed-precision training has not been studied for FNO (Zhao et al., 2022; De Sa et al., 2018; Micikevicius et al., 2017; Jia et al., 2018). The most similar work to ours is FourCastNet (Pathak et al., 2022), a large-scale climate model that uses mixed precision with Adaptive Fourier Neural Operators (Guibas et al., 2021). However, mixed precision is not applied to the FFT or complex-valued multiplication operations, a key challenge the current work addresses. Very recently, another work studies the method of quantization for FNO at inference time (Dool et al., 2023). However, unlike our work, they only study methods that improve memory usage at inference time, not training time. Another paper has appeared recently, which proposes to apply mixed precision to PINNs and DeepONet (Hayford et al., 2024).

## 3 GUARANTEED APPROXIMATION BOUNDS

In this section, to motivate our mixed-precision neural operator training routine in Section 4, we theoretically show that the inherent *discretization error* in the FNO block, from computing the discrete Fourier transform instead of the Fourier transform, is comparable to the *precision error*, from computing the FNO block in half precision rather than in full precision. We present our results in terms of a forward Fourier transform, and we give the full details and discussion in Appendix A.

We start by formally defining the discretization error of the FNO block. Let $D$ denote the closed unit cube $[0,1]^d$ for dimension $d \in \mathbb{N}$, subdivided into $n = m^d$ cubes $Q_1, \ldots, Q_n$ with sidelength $1/m$ and for each $j$, let $\xi_j$ denote the point with the minimum value in each dimension, in $Q_j$. Let $v : D \to D$ denote an intermediate function within the FNO. Recall from the previous section that the primary operation in FNO is the Fourier convolution operator, $(\mathcal{K}v_t)(x)$. We say the *discretization error* of $\mathcal{F}(v)$ is the absolute difference between the Fourier transform of $v$, and the discrete Fourier transform of $v$ via discretization of $v$ on $\mathcal{Q} = (\{Q_1, \ldots, Q_n\}, \{\xi_1, \ldots, \xi_n\})$. Formally, given Fourier basis function $\varphi_\omega(x) = e^{2\pi i \langle \omega, x \rangle}$,

$$\texttt{Disc}(v, \mathcal{Q}, \omega) = \Big| \int_D v(x)\varphi_\omega(x)dx - \sum_{j=1}^n v(\xi_j)\varphi_\omega(\xi_j)|Q_j| \Big|. \tag{1}$$

In other words, if we knew the true (infinite-dimensional) input function, we could run FNO using the (continuous) Fourier transform. However, for real-world applications, we only have access to a discretization of the input, for example, on a $128 \times 128$ grid, so we incur a discretization error. We bound the discretization error as follows. For the full proof details, see Appendix A.

**Theorem 3.1.** *For any $M > 0$ and $L \geq 1$ let $\mathcal{K} \subset C(D)$ be the set of L-Lipschitz functions, bounded by $||v||_\infty \leq M$. Then for all $n, \mathcal{Q}$, there exist $\omega, c_1, c_2 > 0$ such that*

$$c_1\sqrt{d} \cdot Mn^{-2/d} \leq \sup_{v \in \mathcal{K}} (\texttt{Disc}(v, \mathcal{Q}, \omega)) \leq c_2\sqrt{d}(|\omega| + L)Mn^{-1/d}.$$

Intuitively, we bound the Riemann sum approximation of the true integral by showing that in the case where the function $v$ is bounded and Lipschitz, then each of the $n$ intervals contributes $n^{-(1+1/d)}$ error, up to parameters of the function. Note that the discretization error upper bound also scales linearly for the frequency $\omega$, although we empirically show in Section 4 that the energy is concentrated in the lower frequency modes. The lower bound is satisfied when $v(x) = x_1 \cdots x_d$. Furthermore, note that given a function $v$, if $n$ is not large enough, the discretization error can become arbitrarily large due to aliasing. For example, the function $v(x) = M \sin(2\pi(n+\omega)x)$ has discretization error $\Omega(M)$.

Next, we say that the *precision error* of $\mathcal{F}(v)$ is the absolute difference between $\mathcal{F}(v)$ and $\overline{\mathcal{F}(v)}$, computing the discrete Fourier transform in half precision. Specifically, we define an $(a_0, \epsilon, T)$-*precision system* as a mapping $q : \mathbb{R} \to S$ for the set $S = \{0\} \cup \{a_0(1+\epsilon)^i\}_{i=0}^T \cup \{-a_i(1+\epsilon)^i\}_{i=0}^T$, such that for all $x \in \mathbb{R}$, $q(x) = \operatorname{argmin}_{y \in S}|x - y|$. This represents a simplified version of the true mapping used by Python from $\mathbb{R}$ to `float32` or `float16` (we give further justification of this definition in Appendix A). Then, we define

$$\texttt{Prec}(v, \mathcal{Q}, q, \omega) = \Big| \sum_i v(\xi_i)\varphi_\omega(\xi_i)|Q_i| - \sum_i q(v(\xi_i))q(\varphi_\omega(\xi_i))|Q_i| \Big|. \tag{2}$$

Now, we bound the precision error as follows.

**Theorem 3.2.** *For any $M > 0$ and $L \geq 1$ let $\mathcal{K} \subset C(D)$ be the set of L-Lipschitz functions, bounded by $||v||_\infty \leq M$. Furthermore let $q$ be an $(a_0, \epsilon, T)$-precision system. Then for all $n, \mathcal{Q}, \omega$, there exists $c > 0$ such that*

$$\sup_{v \in \mathcal{K}} (Prec(v, \mathcal{Q}, q, \omega)) \leq c \cdot \epsilon M.$$

Once again, we show that each interval contributes $\epsilon/n$ error up to the function's parameters. Taken together, Theorem 3.1 and Theorem 3.2 show that asymptotically, the discretization error can be as large as $Mn^{-2/d}$ (up to constants), while the precision error is always bounded by $\epsilon M$. For example, for `float16` precision ($\epsilon = 10^{-4}$), our theory suggests that the precision error is comparable to the discretization error for three-dimensional meshes up to size $1\,000\,000$. Given this motivation, we present our mixed-precision training pipeline in the next section.

## 4    EMPIRICAL STUDY OF OUR MIXED-PRECISION NEURAL OPERATOR

We empirically study our proposed mixed-precision pipeline for FNO training and demonstrate significant memory usage reduction and large increases in training throughput on various GPUs. We then study failure modes of a naïve application of mixed precision and present our empirically justified pre-activation-based stabilizer solution. Finally, we perform extensive experiments and ablations on our end-to-end training method across different architectures and datasets.

### 4.1    DATASETS AND EXPERIMENTAL SETUP

We summarize each of the four datasets we use; see Appendix B.2 for detailed descriptions.

**Navier-Stokes.** We consider the Navier-Stokes equations for a viscous, incompressible fluid in vorticity form on the unit torus. We use the same dataset as Kossaifi et al. (2023), with a Reynolds number of $500$, composed of $10\,000$ training samples and $2000$ test samples with resolution $128 \times 128$. **Darcy Flow.** We consider the steady-state 2D Darcy Flow equation, which models fluid flow through a porous medium. We use the same dataset as Li et al. (2021a), with 5000 training samples and 1000 test samples at resolution $128 \times 128$. **Spherical Shallow Water Equations.** We use the dataset from Bonev et al. (2023), which generates random initial conditions on the sphere at resolution $256 \times 512$. At each epoch, 120 training samples and 20 validation samples are generated on the fly. **Shape-Net Car and Ahmed-body.** Our final two datasets are 3D real-world car dataset generated by prior work (Umetani & Bickel, 2018; Li et al., 2023), which consists of mesh points that represent a unique 3D car, and the goal is to predict the full 3D pressure field. We use 611 water-tight shapes from car surfaces from Shape-Net (Chang et al., 2015), with 500 samples for training and the rest for the test set. For Ahmed-body, we have 500 for training and 51 for test. The spatial resolution for both is $64 \times 64 \times 64$.

**Experimental Setup.** We run the Navier Stokes and Darcy flow experiments on the TFNO architecture (Kossaifi et al., 2023). For the Spherical SWE, we use SFNO (Bonev et al., 2023) to handle

the spherical geometry, and for Shape-Net Car and Ahmed-body, we use GINO (Li et al., 2023) to handle the large-scale, irregular geometry. All models have the FNO as backbone. We use the official implementation and default hyperparameters for all models.

## 4.2 Mixed-Precision Module for Complex-Valued Tensor Contraction

Now we introduce our theoretically and empirically motivated mixed-precision pipeline for the FNO block. We start by profiling FNO training workloads, identifying the complex-valued tensor contraction within the FNO block as the computational bottleneck, accounting for 4 out of the 5 most time-consuming GPU kernels in the entire training pipeline (see Appendix B.4 for the full profiling results). Furthermore, existing mixed-precision tools such as Pytorch's Automatic Mixed Precision (AMP) (Paszke et al., 2019) leave the operator in full precision since it only autocasts real-valued modules in FNO.

We introduce a simple and lightweight workaround: we break down each tensor contraction into sub-expressions

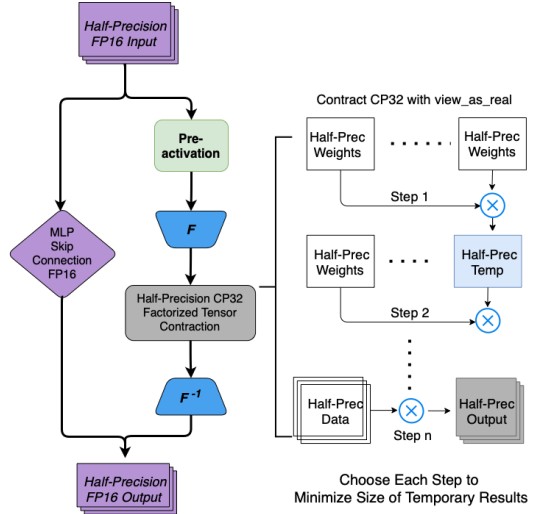

Figure 2: **Overview of our Mixed-FNO pipeline.**

consisting of at most two terms and perform each of these contractions by temporarily converting tensors to reals. To optimize the memory usage, we use a simple greedy algorithm to select the next `einsum` step that minimizes the intermediate tensor size (see Figure 2). The complex-valued operations, including the forward FFT, tensor contraction, and inverse FFT, are done in half-precision. This completes the half-precision FNO block as AMP manages non-FNO, real-valued operations.

Our simple half-precision FNO block module achieves up to 38.6% reduction in memory and up to 50% reduction in memory when combined with Pytorch's native AMP; see Figure 3. Note that the reduction from AMP + Half-Prec FNO is greater than the sum of its parts due to lack of the additional casting to and from half-precision. The memory usage reduction can then be used to train on a larger batch size, thereby improving the overall training throughput and efficiency. On three different Nvidia GPUs, RTX 3090 Ti, V100, and RTX A6000, we demonstrate a consistent improvement in training throughput, in terms of numbers of training samples processed per second, from 1.23X to 1.58X over the baseline in Figure 4.

## 4.3 Numerical Stability via Pre-Activation

Mixed-precision training is prone to underflow and overflow because its dynamic range is significantly smaller than that of full precision (Rakka et al., 2022). Notably, for all four of the datasets in our study, naïvely running FNO in mixed precision results in training failure due to `NaN` outputs. Furthermore, we empirically show that many common solutions, including loss scaling, gradient clipping, normalization, and delaying updates, all fail to address the numerical instability of mixed-precision FNO (see Appendix B.6). To mitigate this overflow issue, we find that pre-activation before each forward FFT is a very effective method for overcoming numerical instability. We also find that the `tanh` pre-activation is the highest-performing operation that we considered according to Appendix B.6. Unlike other functions, `tanh` minimizes changes to small inputs, as it is approximately the identity function near 0 and is smoothly differentiable. Furthermore, `tanh` preserves the discretization-convergent property of FNO. The strong performance of `tanh` is also predicted by our theoretical results in Section 3, since it decreases the $L_\infty$ norm and the Lipschitz constant of the input function. In Appendix B.6, we further show that the `tanh` pre-activation minimally alters the frequency-domain signal in both amplitude and phase. Finally, we demonstrate through an ablation that `tanh` has negligible impact on the model's final error (See Appendix Table 5).

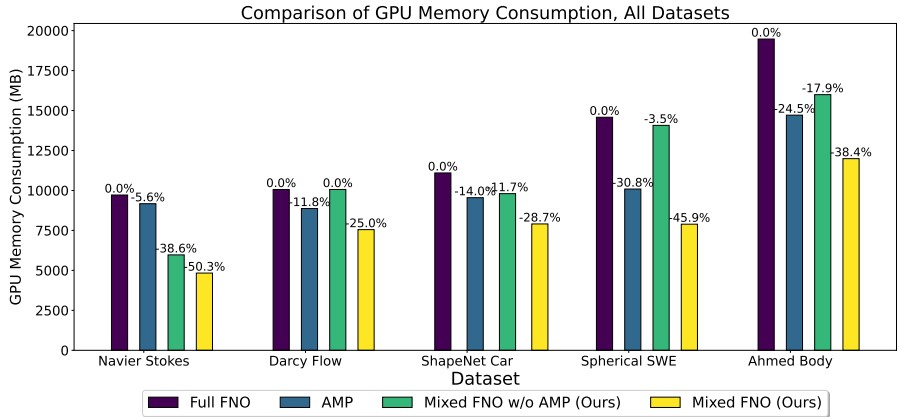

Figure 3: **GPU memory usage reduction across different variants of neural operators on diverse tasks.** We use an Nvidia RTX 3090 Ti GPU. Our method reduces memory by up to 50%, representing a super-linear combination of the two other methods because it avoids additional casting to and from full precision during the forward pass.

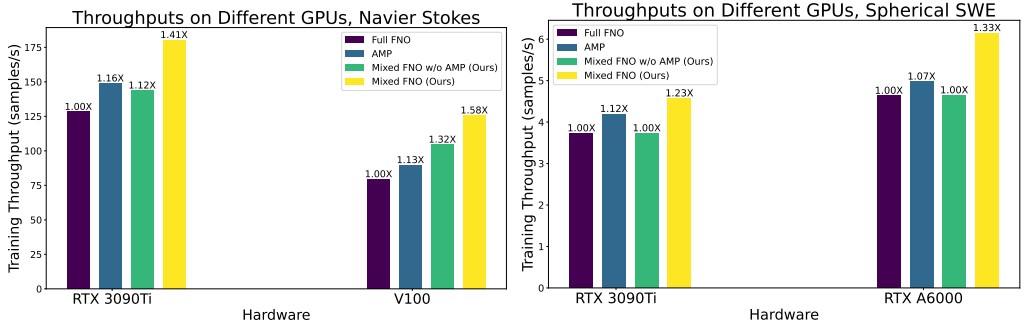

Figure 4: **Training throughput and runtime as a function of the method, on different GPUs.** For mixed-precision FNO + AMP, we consistently observe an improvement of training throughput up to **1.58X** over the baseline with the TFNO model on Navier Stokes, and up to **1.33X** with the SFNO model on Spherical Shallow Water Equations (SWE). Our method also improves upon using only AMP in throughput by over **1.3X** on Navier-Stokes and by over **1.2X** on Spherical SWE. Batch sizes are selected to fully utilize each GPU.

## 4.4 ERROR COMPARISON WITH FULL-PRECISION

Having resolved the critical issue of numerical stability, we demonstrate that our mixed-precision approach achieves errors within 1% of the full-precision baseline across the four datasets. Additionally, we propose a precision-scheduling technique that transitions from mixed to full precision during training, which performs better than full precision in zero-shot super-resolution inference.

**Mixed- vs. Full-Precision Training Curves.** Figure 5 illustrates that our mixed-precision approach achieves test errors on par with full precision throughout the training process, remaining within 1% of the full-precision baseline. We ensure apples-to-apples comparison by keeping all hyperparameters constant across the two precision settings. These hyperparameter configurations are originally optimized for full precision training, so we show that they are directly transferable to mixed precision.

**Precision Scheduling and Zero-Shot Inference.** An important property of neural operators is their *discretization convergence*, meaning that they can be trained on one resolution and tested on a higher resolution (zero-shot super-resolution) (Kovachki et al., 2023). To achieve the best result, we propose a precision schedule, in which the first 25% of training is in mixed-precision, the middle 50% applying only AMP, and the final 25% in full precision. This follows a simple intuition: in the early stages of training, it is okay for gradient updates to be coarser, since the gradient updates are larger overall. However, in the later stages of training, the average gradient updates are much smaller, so

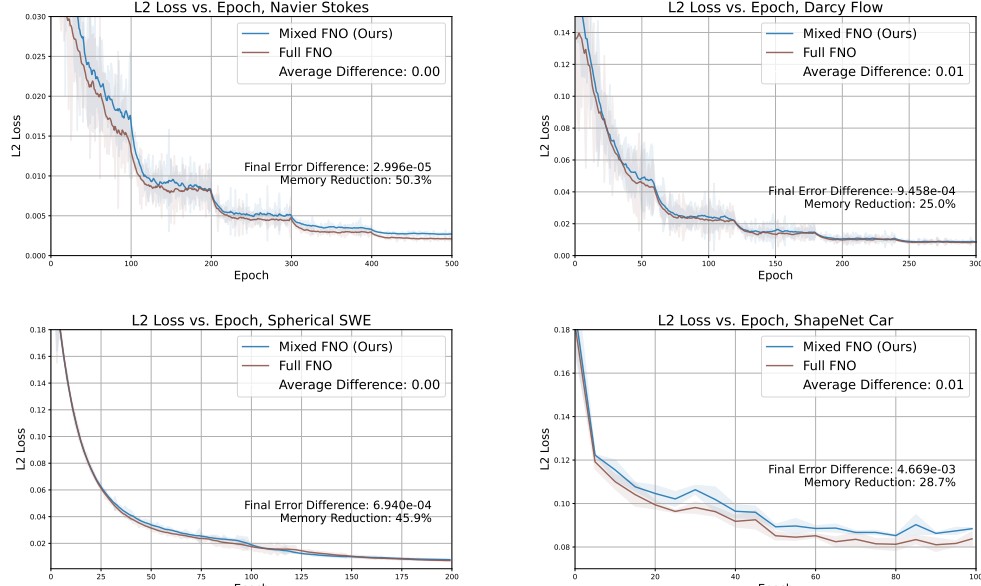

Figure 5: **Test H1 error curves for FNO on the Navier-Stokes (top left) and Darcy flow (top right) datasets.** Test L2 error curves for GINO on the Shape-Net Car (bottom left) and for SFNO on the Shallow Water Equation (bottom right) datasets. Each plot shows the mean of **three** random seeds and standard deviation as error bars. We compute and report the average difference between training curves in the legend. We also annotate the difference in final test errors with the memory savings of our mixed precision approach.

|  | 128x128 | | 256x256 | | 512x512 | | 1024x1024 | |
|---|---|---|---|---|---|---|---|---|
|  | $H^1$ | $L^2$ | $H^1$ | $L^2$ | $H^1$ | $L^2$ | $H^1$ | $L^2$ |
| Full FNO | 0.00557 | 0.00213 | 0.00597 | 0.00213 | 0.00610 | 0.00213 | 0.00616 | 0.00213 |
| Mixed FNO (Ours) | 0.00624 | 0.00236 | 0.00672 | 0.00228 | 0.00688 | 0.00226 | 0.00693 | 0.00226 |
| Precision schedule (Ours) | **0.00503** | **0.00170** | **0.00542** | **0.00170** | **0.00555** | **0.00170** | **0.00558** | **0.00170** |

Table 1: **Zero-shot super resolution**. We test zero-shot super-resolution by training each model on $128 \times 128$ resolution for 19 hours. Mixed precision has a small decrease in accuracy compared to full precision, and using a precision schedule achieves significantly better accuracy.

full precision is more important. We use the Navier-Stokes dataset, trained on $128 \times 128$ resolution (the same setting and model as Figure 5), and tested on $256 \times 256$, $512 \times 512$, and $1024 \times 1024$ resolutions; see Table 1. We find that half-precision has a small decrease in accuracy compared to full precision, and using a precision schedule achieves significantly better generalization.

## 4.5 COMPARISON AGAINST U-NETS

Despite being orignally designed for computer vision tasks, U-Nets have recently been used as PDE surrogates (Ronneberger et al., 2015). Here, we compare FNOs against the U-Net baseline on the Darcy Flow and Navier-Stokes datasets. As shown in Table 2, FNO outperforms UNet: our mixed-precision approach yields higher memory reduction compared to AMP applied to U-Nets.

## 4.6 ABLATION STUDIES

Here, we perform ablations of our mixed-precision procedure on different parameterizations of FNOs, regularization via reducing frequency modes, and training with other numerical systems.

**Decomposition of FNO Weights.** On the Navier-Stokes and Darcy Flow datasets, we adopt a Canonical-Polyadic (**CP**) factorization of the FNO weights using Kossaifi et al. (2019) to ensure better final error. For a generic problem, FNO weights are usually not factorzied and are saved in

| Model | Navier-Stokes | | Darcy Flow | |
|---|---|---|---|---|
| | Error | Memory Reduction | Error | Memory Reduction |
| Full FNO | **0.003** | 50.4% | 0.01 | 25.8% |
| Mixed FNO (Ours) | 0.004 | | **0.007** | |
| Full U-Net | 0.111 | 20.9% | 0.024 | 24.9% |
| U-Net + AMP | 0.111 | | 0.022 | |

Table 2: **Comparison of $L^2$ error and memory reduction with U-Nets**. FNO consistently shows better final error, and our mixed-precision approach yields significantly more memory reduction on Navier-Stokes than AMP applied to U-Nets.

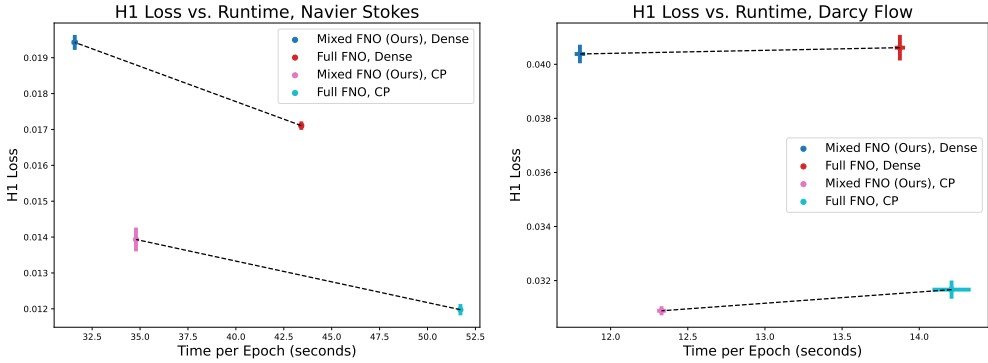

Figure 6: For Navier-Stokes (Left) and Darcy Flow (Right), our method accelerates training, whether with CP factorized weights or Dense unfactorized weights, without losing more than 1% in error.

their original, **dense** form. As shown in Figure 6, we experiment with both scenarios and show that our mixed-precision approach improves runtime without sacrificing accuracy.

**Number of Frequency Modes.** Recall that in the FNO architecture, after the FFT, we truncate to a fixed fraction of frequency modes to improve generalization, typically $1/3$ to $2/3$. We run an ablation study on the number of frequency modes used in the FNO architecture. We run frequency modes $\{16, 32, 64, 128\}$ on the Darcy flow dataset in full and half precision; see Figure 14. We find that using two few frequency modes hurts accuracy substantially while using too many frequency modes increases runtime substantially. There is not a significant difference between half-precision and full precision for all frequencies. In addition, we show in Appendix B.10 that for synthetic data, the precision error from half precision is higher for higher frequencies relative to the amplitude.

**Other Mixed Precision Options.** We also experimented with BrainFloat16 (BF16) and Tensor-Float32 (TF32). However, PyTorch does not support BF16 for discrete Fourier transforms, which are essential to FNOs. Even when applied to the rest of the network, BF16 suffers from error degradation on Navier Stokes, possibly due to having fewer precision bits than FP16. On the other hand, TF32 is not as efficient as our approach, even on optimized hardware such as the A100 GPU. Moreover, we simulated FP8 training via clipping. See Appendix B.11 for additional details.

## 5 CONCLUSIONS AND FUTURE WORK

In this work, we introduced the first mixed-precision training method for neural operators. We derived approximation bounds for mixed-precision neural operator and rigorously characterized the approximation and precision errors of FNO-based operators, proving that the precision error is comparable to the approximation error. Using this insight, we show, in practice, a significant reduction of memory usage and improvement in throughput, without sacrificing accuracy, solving critical issues from standard neural net-based training methods. Through extensive experiments on different state-of-the-art neural operators, datasets, and hardware, we demonstrate that our approach reduces GPU memory usage by up to 50% with little or no reduction in accuracy.

Overall, half-precision FNO makes it possible to train on significantly larger data points with the same batch size. Going forward, we plan to apply this to real-world applications that require super-resolution to enable larger-scale training.

ACKNOWLEDGMENTS

AA is supported by the Bren Foundation and the Schmidt Sciences through the AI 2050 senior fellow program. This project and GP were supported, in part, by the Canada Foundation for Innovation JELF grant, NSERC Discovery grant, AWS Machine Learning Research Award (MLRA), Facebook Faculty Research Award, Google Scholar Research Award, and VMware Early Career Faculty Grant. We would like to express our sincere thanks to Nikola Kovachki for insightful suggestions and discussions, which were invaluable for completing this work. We thank members of the EcoSystem lab, especially Shang Wang, for their constructive feedback.

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

# A  ADDITIONAL DETAILS FROM SECTION 3

In this section, we give the full proofs, details, and discussions from Section 3.

First, for convenience, we restate the definition of the discretization error. Let $D$ denote the closed unit cube $[0, 1]^d$ for dimension $d \in \mathbb{N}$, subdivided into $n = m^d$ cubes $Q_1, \ldots, Q_n$ with sidelength $1/m$. For each $j$, let $\xi_j$ denote the point with the minimum value in each dimension, in $Q_j$. Let $v : D \to D$ denote an intermediate function within the FNO. Recall from the previous section that the primary operation in FNO is the Fourier convolution operator, $(\mathcal{K}v_t)(x)$. We say the *discretization error* of $\mathcal{F}(v)$ is the absolute difference between the Fourier transform of $v$, and the discrete Fourier transform of $v$ via discretization of $v$ on $\mathcal{Q} = (\{Q_1, \ldots, Q_n\}, \{\xi_1, \ldots, \xi_n\})$. Formally, given Fourier basis function $\varphi_\omega(x) = e^{2\pi i \langle \omega, x \rangle}$,

$$\texttt{Disc}(v, \mathcal{Q}, \omega) = \Big| \int_D v(x) \varphi_\omega(x) dx - \sum_{j=1}^n v(\xi_j) \varphi_\omega(\xi_j) |Q_j| \Big|.$$

## A.1  DETAILS FOR RESOLUTION AND PRECISION BOUNDS

In this section, we give the full details for Theorem 3.1 and Theorem 3.2.

**Theorem 3.1 (restated).**  *For any $M > 0$ and $L \geq 1$ let $\mathcal{K} \subset C(D)$ be the set of L-Lipschitz functions, bounded by $\|v\|_\infty \leq M$. Then for all $n, \mathcal{Q}$, there exist $\omega, c_1, c_2 > 0$ such that*

$$c_1 \sqrt{d} \cdot M n^{-2/d} \leq \sup_{v \in \mathcal{K}} (\texttt{Disc}(v, \mathcal{Q}, \omega)) \leq c_2 \sqrt{d}(|\omega| + L) M n^{-1/d}.$$

*Proof.* Let $\varphi_\omega^r(x) = \sin(2\pi \langle \omega, x \rangle)$ denote the real part of the Fourier base of $v$ at frequency $\omega$. Define

$$\texttt{Disc}^r(v, Q, \omega) = \left| \int_D v(x) \varphi_\omega^r(x) dx - \sum_{j=1}^n v(\xi_j) \varphi_\omega^r(\xi_j) |Q_j| \right|. \tag{3}$$

Recall that as defined above, $\mathcal{Q} = (\{Q_1, \ldots, Q_n\}, \{\xi_1, \ldots, \xi_n\})$, where the $n = m^d$ hypercubes of side length $1/m$ subdivide $D$. First we prove the upper bound. We have:

$$\texttt{Disc}^r(v, Q, \omega) = \left| \int_D v(x)\varphi_\omega^r(x)dx - \sum_{j=1}^n v(\xi_j)\varphi_\omega^r(\xi_j)|Q_j| \right| \tag{4}$$

$$= \left| \int_D v(x)\varphi_\omega^r(x)dx - \sum_{j=1}^n \int_{Q_j} v(\xi_j)\varphi_\omega^r(\xi_j)dx \right| \tag{5}$$

$$= \left| \sum_{j=1}^n \int_{Q_j} \left( v(x)\varphi_\omega^r(x) - v(\xi_j)\varphi_\omega^r(\xi_j) \right) dx \right| \tag{6}$$

$$= \left| \sum_{j=1}^n \int_{Q_j} \left( v(x)(\varphi_\omega^r(x) - \varphi_\omega^r(\xi_j)) + (v(x) - v(\xi_j))\varphi_\omega^r(\xi_j) \right) dx \right| \tag{7}$$

$$\leq \sum_{j=1}^n \int_{Q_j} \left( |v(x)| \cdot |\varphi_\omega^r(x) - \varphi_\omega^r(\xi_j)| + |v(x) - v(\xi_j)| \cdot |\varphi_\omega^r(\xi_j)| \right) dx \tag{8}$$

$$\leq \sum_{j=1}^n \int_{Q_j} \left( M \cdot \left( |\omega| \frac{\sqrt{d}}{m} \right) + \left( L \cdot \frac{\sqrt{d}}{m} \right) \cdot 1 \right) dx \tag{9}$$

$$= \sum_{j=1}^n \int_{Q_j} \left( \frac{\sqrt{d}}{m} \left( M|\omega| + L \right) \right) dx \tag{10}$$

$$= n \cdot \left( \frac{1}{m^d} \right) \sqrt{d} \left( M|\omega| + L \right) n^{-1/d} \tag{11}$$

$$= \sqrt{d} \left( M|\omega| + L \right) n^{-1/d}. \tag{12}$$

Bounding $\texttt{Disc}^c(v, Q, \omega)$, the complex part of the Fourier base, follows an identical argument. Setting $c = 2$ concludes the proof of the upper bound, which is true for all $\omega$.

For the lower bound, recall that $D = [0, 1]^d$ and that for each $j$, $\xi_j$ is set to the minimum value in each dimension, in $Q_j$. Setting $v(x) = x_1 \cdots x_d$ and $\omega = 1$, we will show that

$$\left| \int_D v(x) \sin(2\pi x)dx - \sum_{j=1}^n v(\xi_j) \sin(2\pi \xi_j) \right| = \frac{d}{3 \cdot 2^d \pi^{d-2}} \cdot n^{1/2d}.$$

First, in one dimension, $\int_0^1 x_1 \sin(2\pi x_1)dx_1 = \frac{1}{2\pi}$. Therefore, for $d$ dimensions, we have

$$\int_D v(x) \sin(2\pi x)dx = \left( \int_0^1 x_1 \sin(2\pi x_1)dx_1 \right)^d = (2\pi)^{-d}.$$

Next, let $\xi_{j,k}$ denote the $k$th component of $\xi_j$, for $1 \leq k \leq d$ and $1 \leq j \leq n$. We have

$$\sum_{j=1}^{n} \left( v(\xi_j) \sin(2\pi\omega\xi_j)|Q_j| \right) = \frac{1}{m^d} \left( \sum_{j=1}^{n} \left( \prod_{k=1}^{d} v(\xi_{j,k}) \sin(2\pi\omega\xi_{j,k}) \right) \right) \tag{13}$$

$$= \frac{1}{m^d} \left( \sum_{j=1}^{m} \frac{j \cdot \sin(2\pi j/m)}{m} \right)^d \tag{14}$$

$$= m^{-2d} \left( \sum_{j=1}^{m} (j \cdot \sin(2\pi j/m)) \right)^d \tag{15}$$

$$= m^{-2d} \left( -\frac{m}{2}\cot\left(\frac{\pi}{m}\right) \right)^d \tag{16}$$

$$\geq 2^{-d} \cdot m^{-d} \left( -\frac{m}{\pi} + \frac{1}{3} \cdot \frac{\pi}{m} \right)^d \tag{17}$$

$$= (2\pi)^{-d} \cdot \frac{(m + \frac{\pi^2}{3m})^d}{m^{-d}} \tag{18}$$

$$\geq (2\pi)^{-d} \left( 1 + d \cdot \frac{\pi^2}{3} \cdot m^{-2} \right). \tag{19}$$

$$\tag{20}$$

Finally, since $n = m^d$, we have

$$\left| (2\pi)^{-d} - (2\pi)^{-d} \left( 1 + d \cdot \frac{\pi^2}{3} \cdot m^{-2} \right) \right| = \frac{d}{3 \cdot 2^d \pi^{d-2}} \cdot n^{1/2d},$$

which concludes the proof. $\qquad\square$

Note that the upper bound is true for all $\omega$, while the lower bound is shown for $\omega = 1$. It is an interesting question for future work to show the lower bound for $\omega > 1$.

Now we bound the precision error of $\mathcal{F}(v)$. Recall from Section 3 that the *precision error* of $\mathcal{F}(v)$ is the absolute difference between $\mathcal{F}(v)$ and $\overline{\mathcal{F}(v)}$, computing the discrete Fourier transform in half precision. Specifically, we define an $(a_0, \epsilon, T)$-*precision system* as a mapping $q : \mathbb{R} \to S$ for the set $\{0\} \cup \{a_0(1+\epsilon)^j\}_{j=0}^{T} \cup \{-a_j(1+\epsilon)^j\}_{j=0}^{T}$, such that for all $x \in \mathbb{R}$, $q(x) = \text{argmin}_{y \in S}|x - y|$.

This represents a simplified version of the true mapping used by Python from $\mathbb{R}$ to float32 or float16. Recall that these datatypes allocate some number of bits for the mantissa and for the exponent. Then, given a real number $x \cdot 2^y$, its value in float32 or float16 would be $(x + \epsilon_1) \cdot 2^{(y+\epsilon_2)} = x \cdot 2^y + \epsilon_2 x \cdot 2^y + \epsilon_1 \cdot 2^y + \epsilon_1\epsilon_2 2^y = (1 + \epsilon_1/x + \epsilon_2)x \cdot 2^y$. Therefore, our definition above is a simplified but reasonable approximation of floating-point arithmetic.

Now, we define

$$\texttt{Prec}(v, \mathcal{Q}, q, \omega) = \left| \sum_j v(\xi_j)\varphi_\omega(\xi_j)|Q_j| - \sum_j q(v(\xi_j))q(\varphi_\omega(\xi_j))|Q_j| \right|.$$

Now we bound the precision error of $\mathcal{F}(v)$.

**Theorem 3.2 (restated).** *For any $M > 0$ and $L \geq 1$ let $\mathcal{K} \subset C(D)$ be the set of L-Lipschitz functions, bounded by $||v||_\infty \leq M$. Furthermore let $q$ be an $(a_0, \epsilon, T)$-precision system. Then for all $n, \mathcal{Q}, \omega$, there exists $c > 0$ such that*

$$\sup_{v \in \mathcal{K}} \left( \texttt{Prec}(v, \mathcal{Q}, q, \omega) \right) \leq c \cdot \epsilon M.$$

*Proof.* Let $\varphi_\omega^r(x) = \sin(2\pi\langle\omega, x\rangle)$ denote the real part of the Fourier base of $v$ at frequency $\omega$. Define

$$\texttt{Prec}^r(v, Q, q, \omega) = \left| \sum_{j=1}^n v(\xi_j)\varphi_\omega^r(\xi_j)|Q_j| - \sum_{j=1}^n q(v(\xi_j))q(\varphi_\omega^r(\xi_j))|Q_j| \right| \tag{21}$$

We prove the upper bound as follows. We have

$$\begin{aligned}
\texttt{Prec}^r(v, Q, q, \omega) &= \left| \sum_{j=1}^n v(\xi_j)\varphi_\omega^r(\xi_j)|Q_j| - \sum_{j=1}^n q(v(\xi_j))q(\varphi_\omega^r(\xi_j))|Q_j| \right| \\
&= \left| \frac{1}{n} \sum_{j=1}^n (v(\xi_j)\varphi_\omega^r(\xi_j) - q(v(\xi_j))q(\varphi_\omega^r(\xi_j))) \right| \\
&\leq \left| \frac{1}{n} \sum_{j=1}^n (v(\xi_j)(\varphi_\omega^r(\xi_j) - q(\varphi_\omega^r(\xi_j))) + (v(\xi_j) - q(v(\xi_j)))q(\varphi_\omega^r(\xi_j))) \right| \\
&\leq \frac{1}{n} \sum_{j=1}^n (|v(\xi_j)| \cdot |\varphi_\omega^r(\xi_j) - q(\varphi_\omega^r(\xi_j))| + |v(\xi_j) - q(v(\xi_j))| \cdot |q(\varphi_\omega^r(\xi_j))|) \\
&\leq \frac{1}{n} \sum_{j=1}^n (M \cdot (\epsilon \cdot 1) + (\epsilon \cdot M) \cdot 1) \\
&= \frac{1}{n} \cdot n (2\epsilon M) \\
&= 2\epsilon M.
\end{aligned}$$

Bounding $\texttt{Prec}^c(v, Q, q, \omega)$, the complex part of the Fourier base, follows an identical argument. Setting $c = 4$ concludes the proof. □

## A.2 GENERAL BOUNDS

Now, we give results similar to Theorem 3.1 and Theorem 3.2, but with a general function $f$, rather than for $\mathcal{F}(v)$, a function $v$ times the Fourier basis function.

**Theorem A.1.** *For any $M > 0$ and $L \geq 1$ let $\mathcal{K} \subset C(D)$ be the set of L-Lipschitz functions, bounded by $||f||_\infty \leq M$. Then for all $n, \mathcal{Q}, \omega$, we have*

$$2^{-d+1} \cdot d \cdot n^{-1/d} \leq \sup_{f \in \mathcal{K}} (\texttt{Disc}(f, \mathcal{Q}, \omega)) \leq L\sqrt{d} \cdot n^{-1/d}.$$

*Proof.* First, we define

$$\texttt{Disc}^r(f, Q, \omega) = \left| \int_D f(x)dx - \sum_{j=1}^n f(\xi_j)|Q_j| \right|. \tag{22}$$

Recall that $\mathcal{Q} = (\{Q_1, \ldots, Q_n\}, \{\xi_1, \ldots, \xi_n\})$, where the $n = m^d$ hypercubes of side length $1/m$ subdivide $D$. Now we prove the upper bound. We have:

$$\texttt{Disc}^r(f, Q, \omega) = \left| \int_D f(x)dx - \sum_{j=1}^n f(\xi_j)|Q_j| \right| \tag{23}$$

$$= \left| \int_D f(x)dx - \sum_{j=1}^n \int_{Q_j} f(\xi_j)dx \right| \tag{24}$$

$$= \left| \sum_{j=1}^n \int_{Q_j} \left( f(x) - f(\xi_j) \right) dx \right| \tag{25}$$

$$\leq \sum_{j=1}^n \int_{Q_j} |f(x) - f(\xi_j)| \, dx \tag{26}$$

$$\leq \sum_{j=1}^n \int_{Q_j} \left( L \cdot \frac{\sqrt{d}}{m} \right) dx \tag{27}$$

$$= n \cdot \frac{1}{m^d} \left( L \cdot \frac{\sqrt{d}}{m} \right) \tag{28}$$

$$= L\sqrt{d} \cdot n^{-1/d} \tag{29}$$

This concludes the proof of the upper bound.

For the lower bound, recall that for each $j$, $\xi_j$ is set to the minimum value in each dimension in $Q_j$. We set $f(x) = x_1 \cdot x_2 \cdots x_d$.

First, in one dimension, $\int_0^1 x_1 dx_1 = \frac{1}{2}$. Therefore, for $d$ dimensions, we have

$$\int_D f(x)dx = \left( \int_0^1 x_1 dx_1 \right)^d = (2)^{-d}.$$

Next, let $\xi_{j,k}$ denote the $k$th component of $\xi_j$, for $1 \leq k \leq d$ and $1 \leq j \leq n$. We have

$$\sum_{j=1}^n (f(\xi_j)|Q_j|) = \frac{1}{m^d} \left( \sum_{j=1}^n \left( \prod_{k=1}^d f(\xi_{j,k}) \right) \right) \tag{30}$$

$$= \frac{1}{m^d} \left( \sum_{j=1}^m \frac{j}{m} \right)^d \tag{31}$$

$$= m^{-2d} \left( \frac{m(m+1)}{2} \right)^d \tag{32}$$

$$= 2^{-d} \left( 1 + \frac{1}{m} \right)^d \tag{33}$$

$$\geq 2^{-d} \left( 1 + 2dm^{-1} \right)^d \tag{34}$$

$$\tag{35}$$

Finally, since $n = m^d$, we have

$$\left| 2^{-d} - 2^{-d} \left( 1 + 2dm^{-1} \right) \right| = 2^{-d+1} \cdot d \cdot n^{1/d},$$

which concludes the proof. $\square$

Now we similarly bound the precision error of $f$.

**Theorem A.2.** *For any $M > 0$ and $L \geq 1$ let $\mathcal{K} \subset C(D)$ be the set of L-Lipschitz functions, bounded by $||f||_\infty \leq M$. Furthermore let $q$ be an $(a_0, \epsilon, T)$-precision system. Then for all $n, \mathcal{Q}, \omega$, there exists $c > 0$ such that*

$$1/4 \cdot \epsilon M \leq \sup_{f \in \mathcal{K}} \left( Prec(f, \mathcal{Q}, \omega) \right) \leq \epsilon M.$$

*Proof.* Define

$$\texttt{Prec}^r(f, Q, q, \omega) = \left| \sum_{j=1}^{n} f(\xi_j)\|Q_j\| - \sum_{j=1}^{n} q(f(\xi_j))|Q_j| \right| \tag{36}$$

We prove the upper bound as follows. We have

$$
\begin{aligned}
\texttt{Prec}^r(f, Q, q, \omega) &= \left| \sum_{j=1}^{n} f(\xi_j)|Q_j| - \sum_{j=1}^{n} q(f(\xi_j))|Q_j| \right| \\
&= \left| \frac{1}{n} \sum_{j=1}^{n} \left( f(\xi_j) - q(f(\xi_j)) \right) \right| \\
&\leq \frac{1}{n} \sum_{j=1}^{n} |f(\xi_j) - q(f(\xi_j))| \\
&\leq \frac{1}{n} \sum_{j=1}^{n} (\epsilon M) \\
&= \epsilon M
\end{aligned}
$$

This concludes the proof of the upper bound.

For the lower bound, given the definition of an $(a_0, \epsilon, T)$-precision system, it follows that there exists $y$ such that $M/2 < y < M$ and $1/2 \cdot \epsilon y < |y - q(y)|$. Then, for the lower bound we set $f(x) = y$, and we have

$$
\begin{aligned}
\texttt{Prec}^r(f, Q, q, \omega) &= \left| \sum_{j=1}^{n} f(\xi_j)|Q_j| - \sum_{j=1}^{n} q(f(\xi_j))|Q_j| \right| \\
&= \left| \frac{1}{n} \sum_{j=1}^{n} \left( f(\xi_j) - q(f(\xi_j)) \right) \right| \\
&\geq \left| \frac{1}{n} \sum_{j=1}^{n} \left( \frac{1}{2} \cdot \epsilon y \right) \right| \\
&= \frac{1}{2} \cdot \epsilon y \\
&\geq \frac{1}{4} \cdot \epsilon M
\end{aligned}
$$

This concludes the proof. $\qquad\square$

## A.3 PLOTTING THEORETICAL BOUNDS

In this section, we run additional experiments by plotting our theoretical bounds alongside the true empirical discretization and resolution errors of the Darcy flow dataset. Specifically, we plot both our bounds assuming a Fourier basis (Theorem 3.1, Theorem 3.2) and our bounds for general functions (Theorem A.1, Theorem A.2) compared to the true Darcy flow dataset, measured after 10 epochs, at the start of the FNO block (just before the forward FFT). The true errors are calculated using the

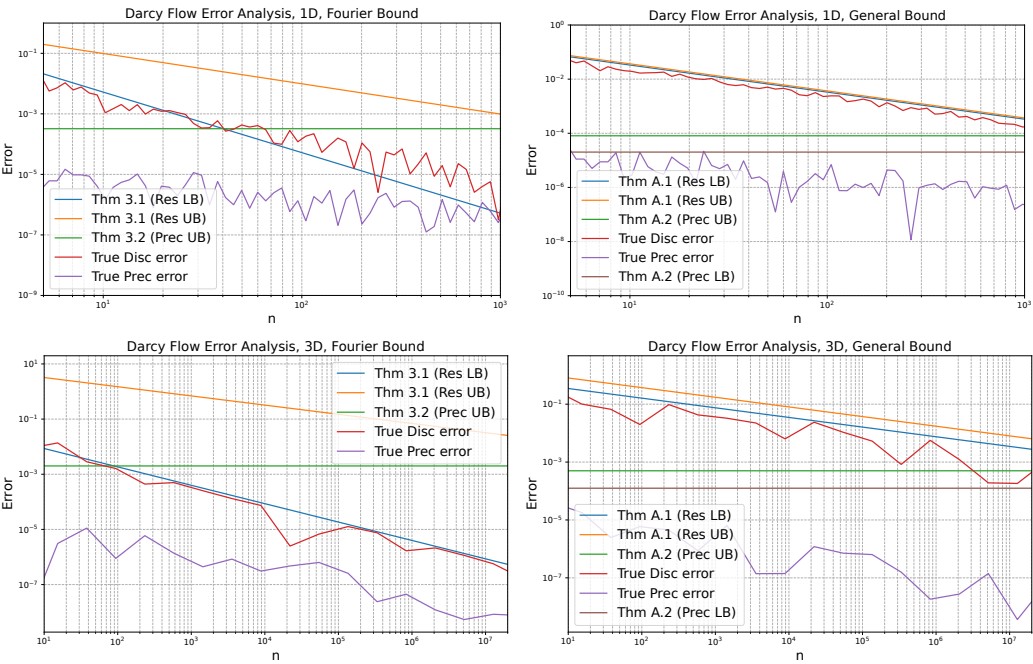

Figure 7: Discretization and precision errors of Darcy flow 1D and 3D, compared to our Fourier basis and general bounds. Note that our bounds are worst-case bounds (we showed there exists a function from the class of bounded $L$-Lipschitz functions which achieves at least the desired error), meaning that the true Darcy flow error may be lower than the worst-case bounds.

definitions of precision and discretization error in the previous section, and we use the true difference in precision between `float32` and `float16` when computing the precision error. See Figure 7. We find that the discretization error is higher than the precision error, as expected by our theory. Furthermore, the Darcy flow discretization and precision errors are always lower than their respective upper and lower bounds. Note that the lower bounds in our theorems are *worst-case* lower bounds (we showed there exists a function from the class of bounded $L$-Lipschitz functions which achieves at least the desired error), which means that the true errors can be lower than the bounds, which is the case in Figure 7.

# B  ADDITIONAL DETAILS FROM SECTION 4

In this section, we give additional details on the datasets and experimental setup, as well as additional experiments.

## B.1  AHMED-BODY CFD EXPERIMENT

Similar to Shape-Net Car, we train the Geometry-Informed Neural Operator on the Ahmed-body dataset, using the same hyperparameters as the original implementation (Li et al., 2023). The memory, error, and throughput results for Ahmed-body are appended to Figure 1. We add the L2 error curves here following the format of Figure 5.

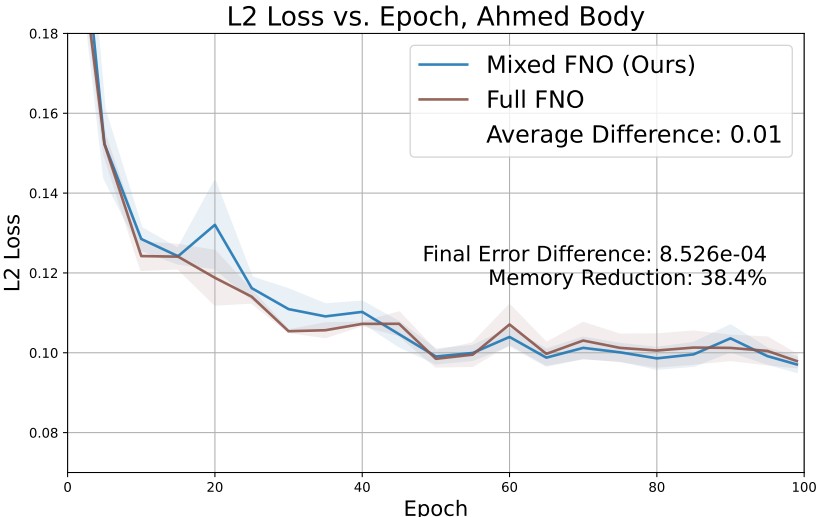

Figure 8: Test L2 error curves for GINO on the Ahmed-body CFD dataset on **three** random seeds and standard deviation as error bars.

## B.2 DATASET DETAILS

**Navier-Stokes.** We consider the 2D Navier-Stokes equation for a viscous, incompressible fluid in vorticity form on the unit torus $\mathbb{T}^2 \cong [0, 2\pi)^2$,

$$\partial_t \omega + \nabla^\perp \phi \cdot \omega = \frac{1}{\text{Re}} \Delta \omega + f, \quad \text{for } x \in \mathbb{T}^2, \ t \in (0, T]$$

$$-\Delta \phi = \omega, \quad \int_{\mathbb{T}^2} \phi = 0, \quad \text{for } x \in \mathbb{T}^2, \ t \in (0, T] \tag{37}$$

where $f \in L^2(\mathbb{T}^2, \mathbb{R})$ is a forcing function and Re $> 0$ is the Reynolds number, for the initial condition $\omega(0, \cdot) = 0$. The goal is to learn the non-linear operator $\mathcal{G}^\dagger : f \mapsto \omega(T, \cdot)$ for $T = 5$ and Re $= 500$. We use the dataset from prior work (Kossaifi et al., 2023), which sets Re $= 500$ and generates forcing functions from the Gaussian measure, $\mathcal{N}(0, 27(-\Delta + 9I)^{-4})$, and computes the solution function via a pseudo-spectral solver (Chandler & Kerswell, 2013). There are 10 000 training and 2000 test samples with resolution $128 \times 128$.

**Darcy Flow.** We consider the steady-state 2D Darcy Flow equation, which models fluid flow through a porous medium. Given $D = (0, 1)^2$, the diffusion coefficient $a \in L^\infty((0, 1)^2; \mathbb{R}_+)$, and the forcing function $f \in L^2((0, 1)^2; \mathbb{R})$, the pressure $u$ satisfies

$$-\nabla \cdot (a(x)\nabla u(x)) = f(x), \quad \text{for } x \in D \tag{38}$$
$$u(x) = 0, \quad \text{for } x \in \partial D \tag{39}$$

We seek to learn the non-linear operator $\mathcal{G}^\dagger : a \mapsto u$, the mapping from the diffusion coefficient $a$ to the solution function $u$. We use the dataset from prior work (Li et al., 2021a), which fixes $f \equiv 1$ and generates 5000 training and 1000 test samples with resolution $128 \times 128$.

**Spherical Shallow Water Equations.** The spherical Shallow Water Equations on the rotating sphere are a model for a variety of geophysical flow phenomena such as atmospheric flows, tsunamis and storm surges. We consider the evolution of two state variables $\varphi, u$ (geopotential height and tangential velocity of the fluid column), governed by the equations

$$\partial_t \varphi + \nabla \cdot (\varphi u) = 0, \quad \text{for } x \in \mathbb{S}^2, t \in (0, T] \tag{40}$$
$$\partial_t (\varphi u) + \nabla \cdot F = S, \quad \text{for } x \in \mathbb{S}^2, t \in (0, T] \tag{41}$$

for initial conditions $\varphi(\cdot, 0) = \varphi_0$, $u(\cdot, 0) = u_0$, where $\mathbb{S}^2$ is the unit sphere, $F$ the momentum flux tensor with $F^{ij} = \varphi u^i u^j + \frac{1}{2}\varphi^2$, and $S$ a source term $S = -2\Omega x \times (\varphi u)$, which models the Coriolis force due to the rotation of the sphere with angular velocity $\Omega$. We use the dataset from prior work (Bonev et al., 2023), which generates random initial conditions on the sphere at resolution $256 \times 512$ and solves the PDE using the spectral solver in the `torch-harmonics` package (Bonev et al., 2023).

**Shape-Net Car**   We evaluate on 3D real-world car dataset generated by prior work (Umetani & Bickel, 2018; Li et al., 2023) using the Reynolds Averaged Navier Stokes (RANS) equations (Wilcox et al., 1998). Each data point consists of mesh points representing a unique 3D car, and the goal is to predict the full 3D pressure field, given an inlet velocity of 20m/s.

**Ahmed-Body CFD**   Including another 3D real-world dataset for evaluation, we the dataset from (Li et al., 2023), which is based on Ahmed-body shapes from (Ahmed et al., 1984), for which steady-state aerodynamic simulations were run using the GPU-accelerated Open-FOAM solver, with varying inlet velocity from 10m/s to 70m/s. The input and output data formats are similar to those of Shape-Net Car, but Ahmed-body has many more mesh points (100k) on the surface.

## B.3   MODEL DETAILS

We use the official implementation of each model, with the default hyperparameters, unless otherwise specified. [1]   Furthermore, we release all of our code at https://github.com/neuraloperator/neuraloperator.

For the case of Shape-Net Car and Ahmed Body on GINO, we note that the default batch size is 1, in fact, the *only allowed* batch size is 1, since each car geometry is unique. As shown in Figure 1, throughput is identical (in fact, our memory saving is enough to have batch size 2, but this is not possible). Although each individual Shape-Net Car fits in memory, other irregular-geometry datasets are too large to fit in memory, so sub-sampling is used (Li et al., 2023). Therefore, mixed-precision GINO allows sampling significantly more mesh points than the full-precision default.

## B.4   FNO PROFILING

Using Nvidia's RTX 3090 Ti, we start by profiling full-precision FNO training on the Navier-Stokes dataset with the PyTorch profiler. Figure 9 provides an overview of runtime breakdown by PyTorch modules and by individual GPU kernels. We observe that complex-valued operations are accountable for the majority of the runtime.

## B.5   NUMERICAL STABILITY: POST-FORWARD METHODS

Most commonly, numerical underflows and overflows that occur during mixed-precision training are handled after the forward pass using loss scaling, gradient clipping, and other methods (Micikevicius et al., 2017). For experimentation, we implemented the following: *(1)* automatic loss scaling via AMP; *(2)* gradient clipping to a default value (5.0); *(3)* delayed updates via gradient accumulation every 3 batches. These methods are compared to our `tanh` approach and a no-stabilizer baseline on the Darcy Flow experiment setting. In Figure 10, we show that all three baselines diverge during the first epoch of training, while `tanh` remains stable numerically. It is noteworthy that while loss scaling almost does not diverge, its scale decreases drastically with each update and becomes infinitesimal. The common problem for global stabilizers is that they do not directly address the numerical overflow of forward FFT operations within each FNO block. In the next subsection, we discuss local methods that acts on the FNO block itself.

## B.6   NUMERICAL STABILITY: PRE-FFT METHODS

Here, we give additional details and experiments for our study on numerical stability of mixed-precision FNO.

---

[1]https://github.com/neuraloperator/neuraloperator

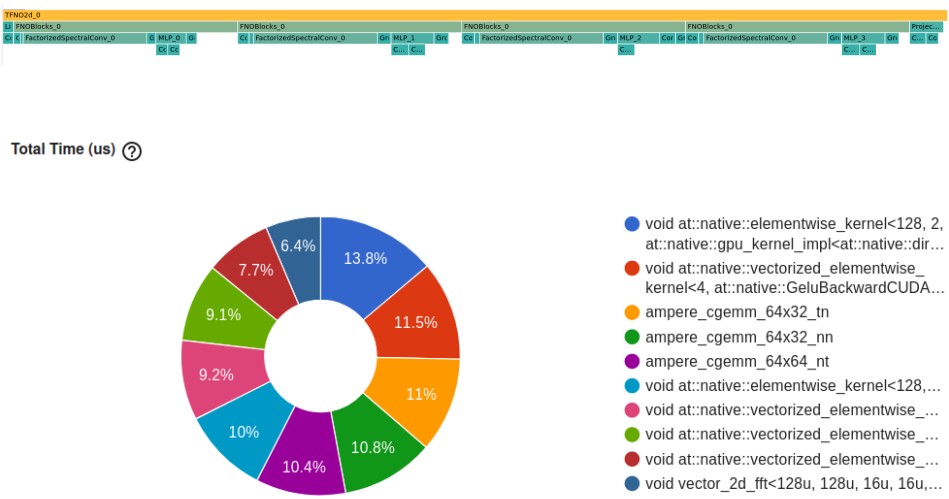

Figure 9: For Navier-Stokes, we use the PyTorch profiler to visualize the runtime breakdown of FNO training by both modules (top) and GPU kernels (bottom). **Spectral convolutions** (complex-valued tensor contraction) accounts for most of the runtime, and correspondingly, kernels associated with complex numbers are 4 amongst the 5 most costly kernels.

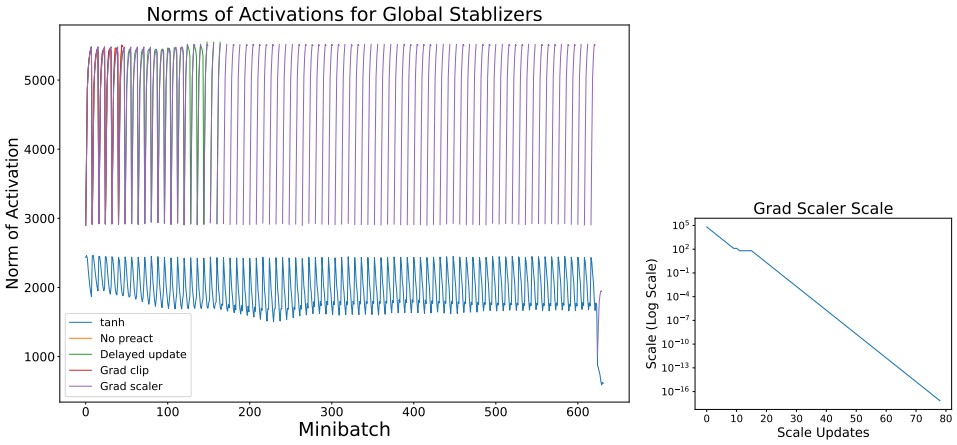

Figure 10: All three global methods diverge during the first epoch of training, with loss scaling completing the most batches without NaN values (Left). However, the scale quickly becomes infinitesimal during training (Right).

A simple idea for addressing overflow in FP16 is to scale down all values by adding a fixed pointwise division operation before the FFT. However, this is an sub-optimal solution because all the values are scaled down equally, removing the data outliers but simultaneously squashing normal data. This forces all numbers into a very small range, which half precision cannot distinguish, preventing the model from converging to an acceptable performance. We find that accuracy deteriorates as the division factor grows: $10^1, 10^2, 10^3$. At $10^4$, the network completely fails to learn due to vanishing gradients. We also find that adding a normalization layer before the Fourier transform (essentially equivalent to scaling) does not fix the numerical instability in the long term. Additionally, normalization layers present extra GPU memory overhead because their statistics have to be tracked for gradient computation.

| Fourier Precision | AMP (T/F) | Pre-Activation | Runtime per epoch | Train loss |
|---|---|---|---|---|
| Full | F | N/A | 44.4 | 0.0457 |
| Full | T | N/A | 42.3 | 0.0457 |
| Half | F | N/A | N/A | N/A |
| Half | T | `hard-clip` | 37.1 | 0.0483 |
| Half | T | $2\sigma$`-clip` | 40.0 | 0.0474 |
| Half | T | `tanh` | 36.5 | 0.0481 |

Table 3: **Comparison of different pre-activation functions used for numerical stability.** `tanh` achieves the fastest runtime without significantly compromising on loss. Since our focus now is on numerical stability, we compare only the average train loss over the 5 final epochs (we compare the test losses in Section 4.4).

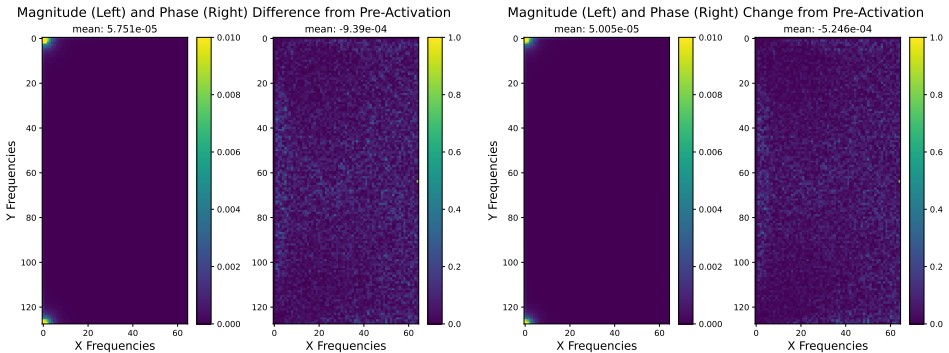

Figure 11: **Impact of pre-activation on frequency domain signals, for a fully-trained full FNO model (Left) and a mixed FNO model (Right),** in terms of mean magnitude and phase differences, on a minibatch of the Navier-Stokes dataset. Pre-activation only changes a extremely small fraction of frequencies, and the two signals are well-aligned in phase.

Appendix B.6 compares several pre-FFT stabilization methods that **can** address numerical instability with acceptable computational costs. `hard-clip` forces maximum and minimum values in the input to a default range. $2\sigma$`-clip` similarly performs clipping but based on the data's mean and standard deviation. Out of these methods, `tanh` is founded to be the highest-performing. After adopting `tanh` as our pre-activation function, we assess its impact on the signal using a trained model, as shown in Figure 11. We find that the loss of signal information is minimal in amplitude and phase.

## B.7 FNO BLOCK PRECISION ABLATION STUDY

As shown in Figure 2, our FNO block has three complex-valued operations that are performed in half-precision: the forward FFT, the tensor contraction, and the inverse FFT. We investigate the effect of altering the precision setting of each on FNO training. This creates a total of eight settings since each operations have two precision configurations: full or half. In Table 4, we show error, runtime, and memory results from training for 30 epochs on the Darcy Flow dataset for each setting. The empirical results demonstrates the acorss-the-board advantage of our approach, which keeps all three operations in half-precision in the forward pass.

## B.8 TANH ABLATION STUDY

A natural question from using pre-activation is to ask how a hyperbolic tangent would affect the full precision FNO model with no other changes. In Table 5, we answer this question by running the full precision FNO model on the Navier-Stokes dataset. We find that there is no noticeable change in the test losses, and the runtime-per-epoch increases by 0.8 seconds, or 1.6%. This concludes that `tanh` is a reasonable and practical choice to improve numerical stability for half precision methods.

| Forward FFT (F/H) | Contraction (F/H) | Inverse FFT (F/H) | Runtime per epoch (sec) | Memory (MB) | Train Error (L2 loss) |
|---|---|---|---|---|---|
| F | F | F | 17.06 | 8870 | 9.00 |
| F | F | H | 16.55 | 8908 | 8.71 |
| F | H | F | 17.11 | 8614 | 8.76 |
| F | H | H | 16.96 | 8658 | 8.15 |
| H | F | F | 17.64 | 7824 | 9.13 |
| H | F | H | 16.81 | 8004 | **7.51** |
| H | H | F | 16.57 | **7558** | 8.75 |
| H | H | H | **15.63** | 7550 | **7.49** |

Table 4: Performance of each setting on Darcy Flow trained for 30 epochs. The fully half-precision setting (our method) is advantageous across all the metrics. Note that the numerical stabilizer is only used when forward FFT is in half-precision.

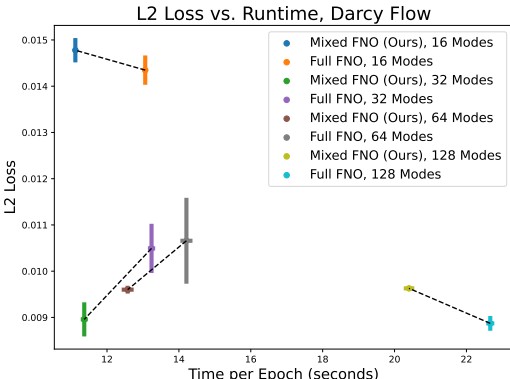

Figure 12: Comparison of the full-precision and mixed-precision (AMP+HALF+TANH) FNO with different frequency modes, on the Darcy flow dataset.

### B.9 COMPARING THE $H^1$ LOSS

We present additional experiments similar to the results in Section 4, but using the $H^1$ metric instead of $L^2$. In Figure 12, we plot an ablation study on the number of frequency modes for Darcy flow, similar to Figure 14 with $L^2$ loss. We see that the overall trends are similar, but test $L^2$ loss is noisier than test L1 loss; this makes sense, because the training loss is $H^1$. In Figure 13, we plot the performance-vs-time and Pareto frontier plots for Navier Stokes and Darcy flow, similar to Figure 5 with $L^2$ loss. As before, we see largely the same trends as with $H^1$ loss, but the results are noisier. In Table 6, we plot the final performance from the models run in Figure 5.

### B.10 FREQUENCY MODE ABLATION STUDY

We run an experiment on synthetic data to demonstrate that the error caused by half precision is higher for higher frequencies, relative to the amplitude. We create a signal based on sine and cosine waves with frequencies from 1 to 10, with randomly drawn, exponentially decaying amplitudes. Then we

Table 5: **Ablation study on full-precision FNO with and without tanh** on the Navier-Stokes dataset. There is no noticeable change in accuracy, showing that tanh is a practical choice to improve numerical stability in low precision methods.

|  | $H^1$ | $L^2$ | time-per-epoch (sec) |
|---|---|---|---|
| Full precision | 0.0121 | 0.00470 | 51.72 |
| Full precision + tanh | 0.0122 | 0.00465 | 52.56 |

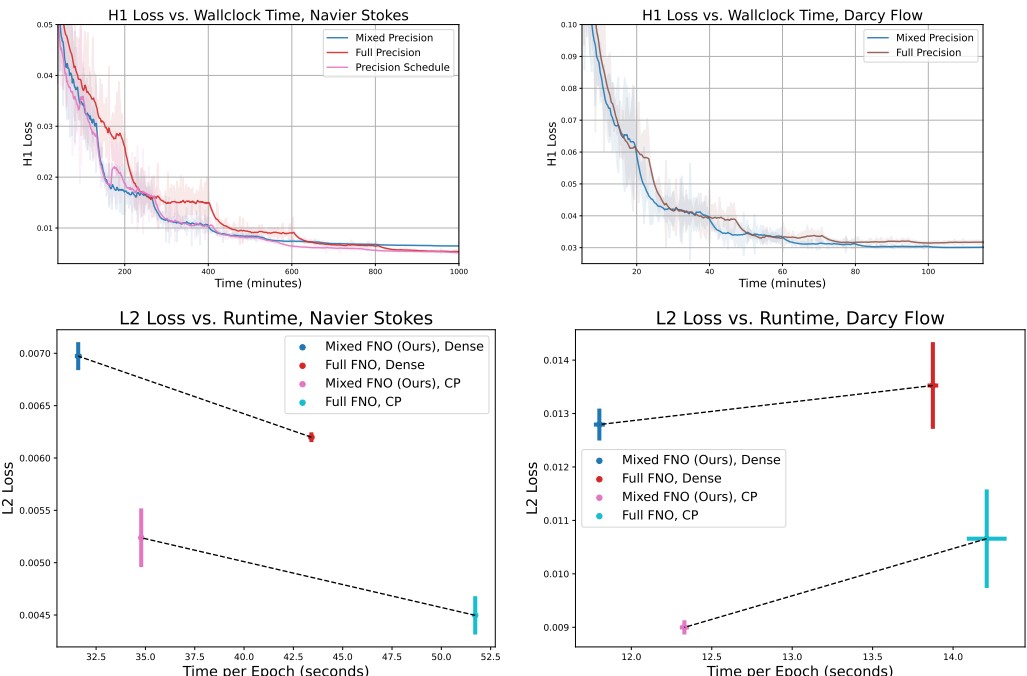

Figure 13: Test $H^1$ error curves for FNO on the Navier-Stokes (top left) and Darcy flow (top right) datasets. Pareto frontier for FNO on the Navier-Stokes (bottom left) and Darcy flow (bottom right) datasets, for Canonical-Polyadic factorization (CP) or no factorization (Dense). We train each model for 500 epochs, and we plot the standard deviation across 3 trials for each setting.

Table 6: Results for FNO with full precision, mixed precision, and precision schedule. All results are the average over 3 seeds trained to 19 hours each (which is the time it takes full precision FNO to reach 500 epochs).

|  | $H^1$ | $L^2$ | time-per-epoch (sec) |
|---|---|---|---|
| Full FNO | $.00536 \pm 2.1e-5$ | $.00214 \pm 3.5e-5$ | 121.4 |
| Mixed FNO (Ours) | $.00645 \pm 6.6e-5$ | $.00212 \pm 1.4e-5$ | 80.2 |
| Precision schedule (Ours) | $.00515 \pm 8.3e-5$ | $.00812 \pm 4.1e-5$ | 80.2, 83.8, 121.4 |

plot the Fourier spectrum in full and half precision, as well as the absolute error of the half-precision spectrum as a percentage of the true amplitudes. See Figure 15; we find that the percentage error exponentially increases. Since in real-world data, the energy is concentrated in the lowest frequency modes, and the higher frequency modes are truncated, this gives further justification for our half precision method.

## B.11 OTHER NUMERIC SYSTEMS

We compare training curves of full FNO, our mixed FNO pipeline, and FNO trained with AMP using Brain Float 16 (BF16) on the Navier-Stokes dataset in Figure 16. In addition to suffering from

|  | Navier-Stokes | Darcy Flow |
|---|---|---|
| FNO + TF32 | 57.44 | 14.12 |
| Mixed FNO (Ours) | **53.72** | **13.52** |

Table 7: **Training time per epoch on a Nvidia A100 GPU.**

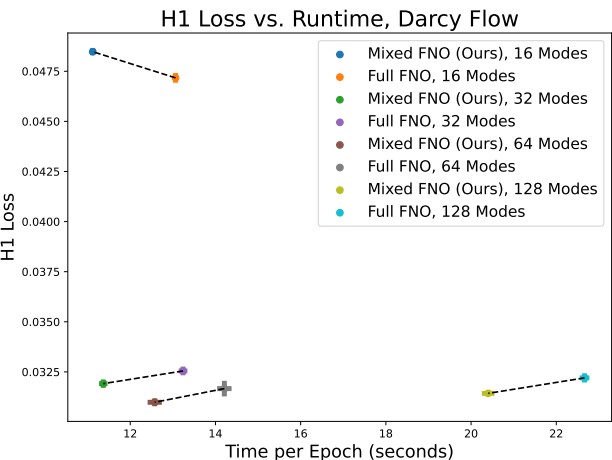

Figure 14: **Comparison of the Full-FNO and Mixed-FNO** with different frequency modes on the Darcy Flow dataset. For 16 frequency modes, the half precision error (compared to full precision) is higher than for 32, 64, or 128 modes.

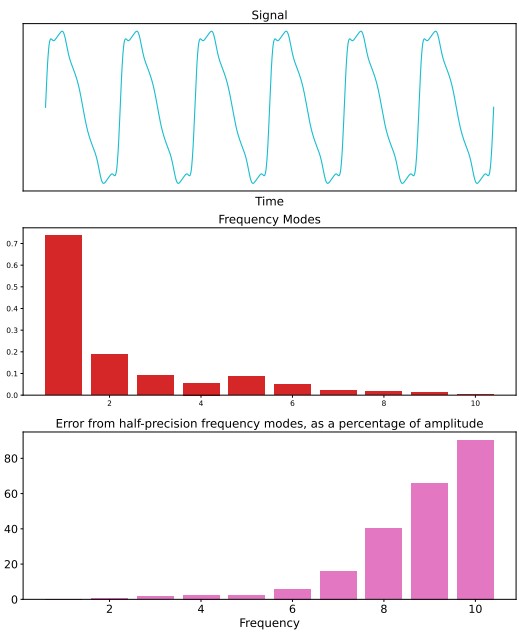

Figure 15: Synthetic signal (top), its frequency modes (middle), and the error due to half precision, as a percentage of the amplitude (bottom). The percentage error increases for higher frequencies.

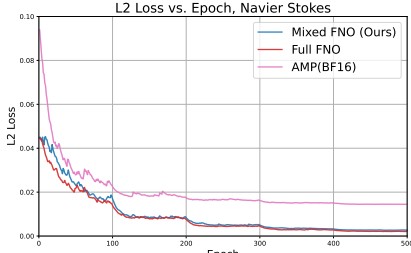 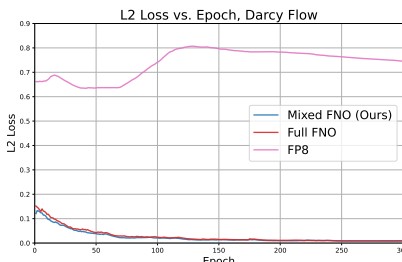

Figure 16: BF16 has significantly higher error compared to the other two approaches, possibly due to reduced bits for precision compared to FP16. FP8 training diverges due to the absence of precision bits in FP16.

reduced precision, BF16 is not supported for most FFT-related operations in PyTorch, which makes its application on FNOs extremely difficult.

Furthermore, we compare the efficiency of our mixed precision approach with TensorFloat 32 (TF32) on an Nvidia A100 GPU, on which TF32 is highly optimized. We report the time per epoch when training on Navier-Stokes and Darcy Flow datasets in Table 7. Our method is still more efficient in comparison.

In addition, we simulate 8-bit floating point (FP8) training using our method on the Darcy Flow dataset via clipping out-of-range values to the maximum and minimum representable under the E5M2 format, which has a higher dynamic range than the E4M3 format (Micikevicius et al., 2022). However, the GINO network diverges during training due to less available precision bits as shown in Figure 16.

Furthermore, the worse result of FP8, is actually predicted Theorem 3.2: while the precision constant for FP16 corresponds to $\epsilon = 10^{-4}$, it corresponds to $\epsilon > 10^{-2}$ for FP8 (even when using E4M3), and we can no longer claim that the precision error is lower than the resolution error.

### B.12 ABLATION STUDY ON CONTRACT AND VIEW-AS-REAL USAGE

This ablation study shows the benefit of (1) decomposition of einsum into sub-equations and appropriate view-as-real usage, (2) our einsum path caching, (3) our greedy tensor contraction procedure, and (4) both weights and inputs in half-precision.

The results are as follows:

1. The usage of view-as-real is different for different tensor contract implementations:
   - Option A: View-as-real on all tensors, and compute a single einsum in one equation (naive);
   - Option B: View-as-real on two tensors at a time, and compute step-by-step einsums with two-tensor sub-equations (optimized);
   - Option C: View-as-real only on high-dimensional einsums, and contracting low-dimension sub-equations in complex format (ours, optimal).

   Results on the Navier-Stokes can be found in Table 8.

2. Re-computing contract paths (naive) vs. Caching contract paths (ours):

   Since tensor shapes are static, we avoid repeated path computation in the default contract implementation. Path computation could take up to 75% of the cost of the contract operation in the forward pass, which we try to avoid.

   Results on the Navier-Stokes and Darcy Flow datasets can be found in Table 9. Our method reduces *time to calculate paths* to almost zero during each iteration of einsum in FNOs.

3. Opt-einsum's FLOP-optimal path (naive) vs. memory-greedy path (ours):

   By default, opt-einsum minimizes FLOPs for the contraction, but our method optimizes memory usage to allow higher-resolution PDEs. Results on Ahmed-body and Shape-Net

Table 8: Comparison of tensor contraction implementations on the Navier-Stokes dataset

| Metric | Option A | Option B | Option C, Ours | Reduction from Option A (%) |
|---|---|---|---|---|
| Time per Epoch (s) | 1730 | 101.7 | **92.59** | 94.65 |
| Memory Usage (MB) | 10310 | 5048 | **4832** | 53.12 |
| Contract Forward Time (s) | 0.178 | 0.0084 | **0.0076** | 95.73 |

Table 9: Re-computing contract paths vs. Caching contract paths

| Dataset | Time to calculate paths (s) | Time to compute einsums (s) | Path/einsum (%) |
|---|---|---|---|
| Navier-Stokes | 0.000573 | 0.000751 | 76.3 |
| Darcy Flow | 0.000441 | 0.000716 | 61.6 |

Car could be found in Table 10. On memory-consuming 3D datasets, our method could save up to 12%.

4. Approximate only weights in half-precision (naive) vs. inputs and weights both half-precision (ours):

If only weights were in half-precision, the memory reduction would have been significantly limited for the contract operation since the inputs are in full-precision. Results on Darcy Flow and Navier-Stokes can be found in Table 11. Note that in the case of Navier-Stokes, pytorch uses a much less memory-efficient kernel if inputs were in full-precision. Hence, it is critical to cast both weights and inputs to half-precision.

Table 10: FLOP-optimal path vs. memory-greedy path

| Dataset | Greedy Memory (MB) | Default Memory (MB) | Reduction (%) |
|---|---|---|---|
| Shape-Net Car | 7906 | 8662 | 8.72 |
| Ahmed-body | 11990 | 13610 | 11.91 |

Table 11: Approximate precision in weights vs. inputs and weights

| Dataset | Half-Prec Memory (MB) | Inputs-Full Memory (MB) | Reduction (%) |
|---|---|---|---|
| Darcy Flow | 7550 | 8166 | 7.54 |
| Navier-Stokes | 4832 | 9380 | 48.46 |

