# OpenReview forum: "Guaranteed Approximation Bounds for Mixed-Precision Neural Operators"
_ICLR.cc/2024/Conference — ICLR 2024 poster_

### Official Review · Reviewer_5SPZ · 2023-10-28

**Soundness:** 3 good
**Presentation:** 3 good
**Contribution:** 2 fair
**Rating:** 6
**Confidence:** 3

**Summary:**

The paper intends to reduce the cost of training and inference of FNO. The method used here include training by mixed-precision to reduce memory and time cost. The mixed-precision training technique proposed here is specialized for FNO, where bounds for precision error and approximation error are theoretically analyzed and guaranteed. The experiment results show that the proposed method greatly improves the speed and throughput of FNO while not sacrificing accuracy.

**Strengths:**

**Originality:** The novelty of this paper lies in leveraging the analysis of precision error in Fourier transform, which successfully reduced the computational cost and maintained the accuracy of FNO.

**Quality:** The quality of the paper is high in both theoretical analysis and experiment conduction.

**Clarity:** The paper delivers its idea in a quite clear and detailed way.

**Significance:** For people who intend to have a large scale of implementations of FNO, this work can be very interesting to them.

**Weaknesses:**

The major doubt for this paper is regarding the significance. While it is dedicated to improving FNO, an important architecture of neural operators, FNO is not the only one and arguably the best one. If FNO is yet to be a dominant model for large scale applications of neural operators, the significance of this work is limited.

**Questions:**

None.

---

> ### Author Response · Authors · 2023-11-17
> **Reply to Reviewer 5SPZ**
>
> We thank you for your thoughtful review. We appreciate that you find our work has high quality theoretical and empirical results, and that our work is novel. Furthermore, we appreciate that you found our work to be clear and detailed. We reply to your questions below.
>
> **Significance.**
> Thank you for motivating this discussion! We give our answer in two parts; first, by motivating the large class of high-performing models that use FNO as a base, and second, by presenting new theoretical results that generalize our original results beyond FNO.
>
> First, our method and findings hold true for any model that is based on FNO, including a large class of state-of-the-art models. Although initially solving classical PDEs, the versatility of FNOs for real-world applications is [gaining recognition](https://www.technologyreview.com/2020/10/30/1011435/ai-fourier-neural-network-cracks-navier-stokes-and-partial-differential-equations/). Our experiments include architectures such as GINO, SFNO that are different from regular FNO and apply to [climate modeling](https://arxiv.org/abs/2306.03838) and [3D car geometries](https://arxiv.org/abs/2309.00583). Recently, new research such as [FourCastNet](https://arxiv.org/abs/2202.11214) have been unveiled that leverages FNO variants and targets high-impact applications, including weather and climate modeling.
>
> Second, we still agree that our work could be even more impactful if we generalize our results. Therefore, we have now added **new theoretical results**, that generalize our original bounds using the Fourier basis, to arbitrary basis functions (Theorems A.1 and A.2 in Appendix A). This extends our results beyond Fourier neural operators to a larger class of integral operators, that can include e.g. Wavelet bases or graph neural operators. We also have initial simulation results which confirm our new theoretical results, in Appendix A.3.
>
> Finally, we would also like to highlight a few other updates we made to our paper
> - We give new experiments that directly compare our theoretical resolution and precision upper and lower bounds, to the empirical discretization and precision error (Appendix A.2). We find that the errors behave as expected by our theory, which is additional confirmation of the correctness of our theory and the applicability of our results.
> - We added another experiment setting, the Ahmed body CFD dataset. On this high-resolution 3D task, we achieve 38.4% memory reduction while the error stays within 0.05% of the full-precision model.
>
> Thank you once again, for your excellent point. Please let us know if you have any follow-up questions or further suggestions. We would be happy to continue the discussion.

---

> > ### Comment · Reviewer_5SPZ · 2023-11-22
> >
> > Thank you for your reply. I have read other reviews and would like to keep my score.

---

### Official Review · Reviewer_NM9S · 2023-10-31

**Soundness:** 3 good
**Presentation:** 4 excellent
**Contribution:** 3 good
**Rating:** 8
**Confidence:** 3

**Summary:**

The authors demonstrated mixed-precision training of FNOs on GPUs.  They derived theoretic bounds on the effect of rounding error, improved AMP to support complex arithmetic, addressed an instability by introducing a pre-activation thah, and conducted experiments on practical datasets.

**Strengths:**

- Application is of significance.
- Well written.
- Code released.

**Weaknesses:**

> It would be helpful to include an illustration of empirical characterization of rounding error in direct comparison with the theoretical bound scaling law.
> Mixed-precision with FP8 has been proposed, how does the method fare with FP8?

**Questions:**

See minor requests above.

---

> ### Author Response · Authors · 2023-11-17
> **Reply to Reviewer NM9S**
>
> Thank you for your positive feedback. We appreciate that you find our work is well-written and covers an application that is significant. We reply to your points below.
>
> **W1: Illustration of empirical characterization.**
> Thank you for the suggestion; this is a great idea! Following your suggestion, we added new experiments to Appendix A.2 that directly compare our theoretical resolution and precision upper and lower bounds, to the empirical resolution and precision error. We find that the discretization error is higher than the precision error, as expected by our theory, which is additional confirmation of the correctness of our theory and the applicability of our results. These experiments also make it easy to see qualitative features of our theoretical results, such as how the discretization error decays as the grid size increases, while the precision error stays constant.
>
> **W2: Mixed-precision with FP8.**
> Thank you, this is another interesting suggestion. Despite there being no native support for FP8 in pytorch, we are able to simulate FP8 training through two approaches:
> 1. Calibrating network activations from FP16 to FP8 using two format options, E5M2 and E4M3. Unfortunately, for both formats, the training instantly fails in the first iteration due to FNO producing inf values. While prior work has used FP8 for GEMM and convolution operations (as in [this paper](https://proceedings.neurips.cc/paper_files/paper/2018/file/335d3d1cd7ef05ec77714a215134914c-Paper.pdf), to the best of our knowledge, we are not aware of any work that computes Fourier transforms in FP8.
> 2. Clipping out-of-range activations to the upper and lower limits of FP8. To mitigate training divergence, we constantly check for out-of-range values and clip them to the extrema of FP8 formats. Although training could proceed as normal, the FP8-FNO is unable to converge as shown in the Darcy Flow training curve comparing FP8 and FP16 training. Training curves are added in Appendix B.11.
>
> Furthermore, the worse result of FP8 is actually predicted by our theory. In our paper, Theorem 3.2 showed that for FP16, precision error is comparable to discretization error when plugging in $\epsilon=10^{-4}$, the multiplicative precision of FP16. However, for FP8 $\epsilon>10^{-2}$, and we can no longer prove that the precision error is lower than the discretization error. We have added this discussion to Appendix B.11.
>
> Finally, we would also like to highlight a few other updates we made to our paper
> - Improved theoretical bounds: while we originally proved bounds using the Fourier basis, we have now proven matching results for arbitrary basis functions (Appendix A.2).
> - We added another experiment setting, the Ahmed body CFD dataset. On this high-resolution 3D task, we achieve 38.4% memory reduction while the error stays within 0.05% of the full-precision model.
>
> Thank you once again, for your favorable assessment of our work and your excellent suggestions. We would be very happy to answer follow-up questions or follow any additional suggestions.

---

### Official Review · Reviewer_dRFB · 2023-10-31

**Soundness:** 3 good
**Presentation:** 3 good
**Contribution:** 2 fair
**Rating:** 5
**Confidence:** 4

**Summary:**

In this paper the authors provide a way to apply mixed-precision training to Fourier Neural Operators. The authors point out that while some previous approaches have used mixed-precision training with FNO, they have done so only on the real-valued parameters (i.e, the linear weights, and biases) and not on the FFT/DFT that is applied to the input in each layer.

They first show that mixed-precision training should not cause extra error in the training due to the Fourier transform, since the discretization error from FFT→ DFT is orders of magnitude bigger than the DFT→ half-precision DFT.

For the application of mixed-precision on complex tensors (the output after an FFT/DFT application) the authors apply mixed-precision by converting each “tensors to real” (this is something that the authors don’t elaborate upon enough).

The authors show through their experiments, that they are indeed able to train various FNO based architectures with low-memory and hence are also able to increase the throughput of their runs while only taking a less than 1 percent hit in the performance as compared to non-mixed-precision baselines.

The authors also point to the use of a tanh based pre-activation that helps in mitigating the mixed-precision based overflow that usually occurs when trained FNO based architectures.

**Strengths:**

The paper is well-written and easy to follow.

The experimental results of the paper are impressive. The authors are able to increase the training throughput for navier stokes $1.41$ times, by achieving an almost 50 percent memory reduction.

The use of tanh activation as a pre-activation to the neural operator block is a good technique to reduce the overflow issue, and an important finding.

The authors provide a theoretical proof for why the error for the mixed-precision training would be negligible when compared to the discretization error for DFT, while the results there are pretty standard its a good addition to have.

**Weaknesses:**

I think that the primary methodology as to how the mixed-precision is applied to the Fourier Kernel is not clear. From the looks of it the mixed-precision approximation is applied to the weights in the complex domain (that are used in the kernal operator).

However, this is something that seems to be different from what they say in the introduction, where they mention that they enable some mixed precision in the entire FNO block (which I assumed will try to also enabled mixed precision in the DFT algorithm).

From the looks of it the primary contribution seems to be the addition of tanh, and the application of mixed precision in the complex domain by treating the real and the imaginary components as two distinct real tensors. Together, the overall contribution does not seem very novel in of itself.

**Questions:**

The authors mention that neural operators are discretization convergent, however have not cited any relevant work around it. While empirically we know that for some FNO based architecture we can get zero shot super-resolution, are there any works that prove it? In general, adding relevant citation to that claim would be useful.

---

> ### Author Response · Authors · 2023-11-17
> **Reply to Reviewer dRFB**
>
> We thank the reviewer for taking the time to review the manuscript and provide valuable feedback. We are glad to see that you found our experimental results to be impressive. We address your questions below.
>
> **W1: Unclear methodology.**
> Thank you for pointing this out. We are happy to clarify that our methodology does run the entire FNO block in half-precision, with the forward FFT, the tensor contraction, and the inverse FFT all in half precision. We have now updated Section 4.2 and Figure 2 to clarify this further.
> Based on your question, we now also added a more extensive ablation study on which FNO block operations to keep in half-precision, which we have added to Appendix B.7. Forward FFT, tensor contraction, and inverse FFT can each be set to each full or half precision, creating a total of 8 settings. Our study shows the half-precision FNO block (our method) achieving better runtime, memory usage, and training error than all 7 other settings (this is consistent with our original Darcy flow results, in which half-precision outperforms full-precision).
>
> The full results are as follows:
> Performance of each setting on Darcy Flow. The fully half-precision setting (our method) is advantageous across all the metrics.
>
> | Forward FFT (F/H) | Contraction (F/H) | Inverse FFT (F/H) | Runtime per epoch (sec) | Memory (MB) | Train Error (L2 loss) |
> |-------------------|-------------------|-------------------|------------------------|-------------|-----------------------|
> | F             	| F             	| F             	| 17.06              	| 8870    	| 9.00              	|
> | F             	| F             	| H             	| 16.55              	| 8908    	| 8.71              	|
> | F             	| H             	| F             	| 17.11              	| 8614    	| 8.76              	|
> | F             	| H             	| H             	| 16.96              	| 8658    	| 8.15              	|
> | H             	| F             	| F             	| 17.64              	| 7824    	| 9.13              	|
> | H             	| F             	| H             	| 16.81              	| 8004    	| **7.51**          	|
> | H             	| H             	| F             	| 16.57              	| **7558**	| 8.75              	|
> | H             	| H             	| H             	| **15.63**          	| **7550**	| **7.49**          	|
>
>
> **W2: On novelty.**
> We respectfully note that our novelty “lies in leveraging the analysis of precision error in Fourier transform, which successfully reduced the computational cost and maintained the accuracy of FNO” (as pointed out by reviewers XXPr, 5SPZ). Second, the goal of our paper is to establish a method that improves the throughput and memory footprint within the domain of scientific machine learning and particularly neural operators. In the literature prior to our work, it was not clear if mixed-precision techniques could be used in this domain, especially since classical numerical solvers traditionally suffer from stability issues and sometimes need **higher** than standard precision. However, our paper shows that numerical stability is quite promising (with up to 50% reduction in memory and throughput) and our work generally lays a theoretical and empirical foundation in this area.
>
> **Q1: discretization convergent.**
> Thank you for this question. Yes, [this paper](https://www.jmlr.org/papers/v24/21-1524.html) gives a formal proof of discretization convergence for neural operators, including FNO (Section 9.2, Theorem 8). We have now added this reference in the paper. This shows that the approximation error is approximately constant as the resolution increases. As you mentioned, we also empirically verify this property via our zero-shot super-resolution experiments (Table 1).
>
> We would also like to highlight a few other updates we made to our paper
> - Improved theoretical bounds: while we originally proved bounds using the Fourier basis, we have now proven matching results for arbitrary basis functions (Appendix A.2).
> - We added another experiment setting, the Ahmed body CFD dataset. On this high-resolution 3D task, we achieve 38.4% memory reduction while the error stays within 0.05% of the full-precision model.
>
> Thank you very much, once again, for your generally positive view of our work and your excellent suggestions. If you find our responses satisfying, we respectfully ask that you consider increasing your score. On the other hand, we would be very happy to answer any follow-up or additional questions you have.

---

> > ### Author Response · Authors · 2023-11-21
> > **Second Reply to Reviewer dRFB**
> >
> > Thank you once again for your great suggestions. Please let us know if you have any follow-up questions to our reply. We would be happy to reply any time before the author response period closes tomorrow (end of day Nov 22nd).
> >
> > We also would like to mention that we have just completed an additional ablation study, to further address your comments about our methodology. In particular, this ablation study shows the benefit of (1) decomposition of einsum into sub-equations and appropriate view_as_real usage, (2) our einsum path caching, (3) our greedy tensor contraction procedure, and (4) both weights and inputs in half-precision. These results complement the results from our first response, as well as the ablation study in our original submission. Please see the results below.
> >
> > (1). The usage of view_as_real is different for different tensor contract implementations:
> > Option A: View_as_real on all tensors, and compute a single einsum in one equation (naive);
> > Option B: View_as_real on two tensors at a time, and compute step-by-step einsums with two-tensor sub-equations (optimized);
> > Option C: View_as_real only on high-dimensional einsums, and contracting low-dimension sub-equations in complex format (ours, optimal).
> >
> > Results on the Navier Stokes dataset:
> > | Metric            | Option A   | Option B        | Option C, Ours      | Reduction from Option A (%) | Reduction from Option B (%) |
> > |-----------------------|--------------|--------------|--------------|------------------------------|--------------------------|
> > | Time per Epoch (s)| 1730         | 101.7        | 92.59        | 94.65                        | 8.95                     |
> > | Memory Usage (MB) | 10310        | 5048         | 4832         | 53.12                        | 4.28                     |
> > | Contract Forward Time (s)  | 0.178        | 0.0084       | 0.0076       | 95.73                        | 9.52                     |
> >
> > (2) Re-computing contract paths (naive) vs. Caching contract paths (ours):
> > Since tensor shapes are static, we avoid repeated path computation in the default contract implementation. Path computation could take up to 75% of the cost of the contract operation in the forward pass, which we avoid via caching.
> >
> > | Dataset | Time to calculate paths (s) | Time to compute einsums (s) | Path/Einsum (%) |
> > |----------|----------------------------|-------------------------------|-----------------|
> > | Navier Stokes  | 0.000573                	| 0.000751                  	| 76.3        	|
> > | Darcy Flow 	| 0.000441                	| 0.000716                  	| 61.6        	|
> >
> > Our caching reduces “time to calculate paths” to almost zero during all but the first iteration of einsum in FNOs.
> >
> > (3)  [Opt_einsum’s](https://github.com/dgasmith/opt_einsum/tree/c826bb7df16f470a69f7bf90598fc27586209d11) FLOP-optimal path (naive) vs. memory-greedy path (ours)
> >
> > By default, opt_einsum minimizes FLOPs for the contraction, but our method optimizes memory usage to allow higher-resolution PDEs.
> >
> > | Dataset | Greedy Memory (MB) | Default Memory (MB) | Reduction (%) |
> > |---------|--------------------|---------------------|---------------|
> > | Car-CFD     | 7906               | 8662                | 8.72          |
> > | Ahmed-body   | 11990              | 13610               | 11.91         |
> >
> > Our method save up to 12% on memory usage.
> >
> > (4) Approximate only weights in half-precision (naive) vs. inputs and weights both half-precision (ours)
> >
> > If only weights were in half-precision, the memory reduction would have been significantly limited for the contract operation.
> >
> > | Dataset | Half-Prec Memory (MB) | Inputs-Full Memory (MB) | Reduction (%) |
> > |---------|----------------------|-------------------------|---------------|
> > | Darcy Flow   | 7550                 | 8166                    | 7.54          |
> > | Navier Stokes  | 4832                 | 9380                    | 48.46         |
> >
> > Note that in the case of Navier Stokes, pytorch uses a much less memory-efficient kernel if inputs are in full-precision. Hence, it is critical to cast both weights and inputs to half-precision.
> >
> > Once again, we look forward to answering any follow-up or additional questions you have before the author response period closes tomorrow. Thank you!

---

### Official Review · Reviewer_XXPr · 2023-11-02

**Soundness:** 3 good
**Presentation:** 4 excellent
**Contribution:** 4 excellent
**Rating:** 8
**Confidence:** 4

**Summary:**

The paper introduces a mixed-precision training method for neural operators, focusing on Fourier Neural Operators (FNO) used in solving partial differential equations and other mappings between function spaces. The paper discusses the challenges of high-resolution training data, limited GPU memory, and long training times in the context of neural operators. It emphasizes the need for mixed-precision training to mitigate these issues. The paper demonstrates that, contrary to our expectations, mixed-precision training in FNO does not lead to significant accuracy degradation. It presents rigorous theoretical characterization of approximation and precision errors in FNO, highlighting that the precision error is comparable to the approximation error. The paper introduces a method for optimizing memory-intensive tensor contractions using mixed precision, reducing GPU memory usage by up to 50% and improving throughput by 58% while maintaining accuracy.

 These findings have the potential to advance the efficiency and scalability of neural operators in various downstream applications.

**Strengths:**

The strengths of the paper are as follows:

1. Mixed-Precision Training for Neural Operators: The paper introduces the first mixed-precision training routine tailored specifically for neural operators. This novel approach optimizes the memory-intensive tensor contraction operations in the spectral domain and incorporates the use of tanh pre-activations to address numerical instability.

2. Theoretical Results; The paper provides a strong theoretical foundation for its work by characterizing the precision and discretization errors of the FNO block. It demonstrates that these errors are comparable and proves that, when executed correctly, mixed-precision training of neural operators does not lead to significant performance degradation.

3. Experimental Evaluation: The paper conducts thorough empirical validation of its mixed-precision training approach on three state-of-the-art neural operators (TFNO, GINO, and SFNO) across four different datasets and GPUs. The results indicate that the method significantly reduces memory usage (using half the memory) and increases training throughput by up to 58% across various GPUs, all while maintaining high accuracy with less than 0.1% reduction.

4. Open-Source Code: The paper provides an efficient implementation of its approach in PyTorch, making it open-source and providing all the necessary data to reproduce the results.

**Weaknesses:**

While the method of using tanh pre-activation before each FFT seems to avoid numerical instability, it would have been better if some theoretical justification was given (even for simplistic cases). I believe similar theory should have been provided for the learning rate schedule. However, the paper indeed contributes significantly in theoretical aspects (in Section 3) by characterizing the precision and discretization errors of the FNO block and showing that these errors are comparable. Given the strong theoretical contributions, the above is not significant. Moreover, a strong ablation study for tanh is provided in Appendix B.5

**Questions:**

The work is indeed very interested and provides a strong contribution to the research community. My questions are:

1. Why is it that the standard mixed precision training used for training ConvNets and ViTs is ineffective for training Fourier Neural Operators?
2. Similarly, why is is that the common solutions (loss scaling, gradient clipping, normalization, delaying updates) fail to address the numerical instability of mixed-precision FNO?

---

> ### Author Response · Authors · 2023-11-17
> **Reply to Reviewer XXPr (1/2)**
>
> Thank you for your favorable and detailed review. We appreciate that you find our work very interesting and provide a strong contribution to the research community. Further, we are glad that you found that our work has the potential to advance the efficiency and scalability of neural operators in downstream applications. We address each of your questions below:
>
> **W1: Justification for pre-activation before each FFT.**
> This is a great question. In fact, pre-activation is predicted by our theory, and we have now updated our paper to point this out. Given a function $v(x)$ defined on $[0,1]$, as long as $|\max_x v(x)-\min_x v(x) |>1$, then $\text{tanh}(v(x))$ has both a lower $L_\infty$ norm and a lower Lipschitz constant compared to $v(x)$, both of which are terms that show up in the upper bounds of Theorem 3.1 and Theorem A.1. In other words, tanh decreases the magnitude and gradient of the functions, which decrease the theoretically-predicted discretization and precision errors. Finally, as you say, we also have an ablation study that empirically motivates tanh in Appendix B.8.
>
> **W2: Justification for learning rate schedule.**
> First, we would like to clarify that you meant the precision schedule (our contribution), instead of the learning rate schedule (well-known in the literature)? The precision schedule follows the following intuition: in the early stages of training, it is okay for gradient updates to be more “coarse”, since the gradient updates are larger overall, so full precision is less important. However, in the later stages of training, the average gradient updates are much smaller, so full precision is important. In fact, we find that at the end stages of training, the magnitudes of some gradients in the FNO block fall below $10^{-4}$, which is below the precision constant of float16. This aligns with the empirical results of our precision schedule experiments: full precision performs well at the last stages of training.  We have now updated our manuscript with this explanation.
>
> **Q1: Standard techniques ineffective for FNO.**
> This is a great question; we have now updated the paper to clarify this. The standard techniques are designed for common operations such as convolution and transformers, however, the most memory-intensive operations of FNO are in the spectral domain, which are not supported by AMP and require a different procedure that takes into account both casting to half precision as well as several vectorization operations (moving to and from complex-valued representation). Since complex-valued operations are not supported in AMP, it simply skips the most important operations inside the FNO block, and therefore only results in a small improvement for FNO. We introduce the first mixed-precision training method for the FNO block by devising a simple and lightweight greedy strategy to find the optimal contraction order, taking into account the vectorization operations for each intermediate tensor. Furthermore, numerical stability methods also must be handled differently for neural operators, since the typical methods fail, which we discuss below in Q2.
>
> (1/2)

---

> ### Author Response · Authors · 2023-11-17
> **Reply to Reviewer XXPr (2/2)**
>
> **Q2. Common numerical stability techniques.**
> Thank you for the question! While the main source of instability in mixed-precision neural nets is the gradient updates, we find that for neural operators, there is even more numerical instability within the forward FFT operations. Therefore, the standard techniques such as loss scaling, gradient clipping, and delayed updates, are focused on reducing the dynamic range of the gradients, but for neural operators, we are interested in addressing the numerical instability of the FFT. We now further verify these claims by significantly extending our original ablation study, which we have also added to Appendix B.5.
> - Loss scaling: we now confirm in Appendix B.5 that the typical loss scaling (GradScaling) is ineffective, due to the reason above. Instead, we tried scaling just before each FFT. However, this fails due to the wide range of values. For example, if we typically have a few outliers of $10^6$, while most of the numbers are within the range $[-1,1]$, then we would expect scaling to perform much worse than tanh, since tanh negates the outliers while keeping the same range in the rest of the values. We confirm this in a new ablation study in Appendix B.5.
> - Gradient clipping: this also does not work due to the above reason. In our new ablation study, we clip gradients to a default value of 5 during training, yet it does not resolve the NaN issue. Instead, we tried two different forms of clipping the activations just before the FFT (from our original submission), in Appendix B.5.
> - Normalization: note that we already do have normalization in our models. In our new ablation study, we added another normalization layer before the FFT, which delays the NaN occurrence only for 2 epochs of training. Additionally, this incurs a 30% GPU memory overhead due to intermediate values stored for the backward pass. Our tanh approach is more lightweight and effective.
> - Delaying updates: for gradient accumulation, we tried updating weights every 3 batches, which did not resolve the NaN problem.
>
> Finally, we would like to highlight a few other updates we made to our paper
> - Improved theoretical bounds: while we originally proved bounds using the Fourier basis, we have now proven matching results for arbitrary basis functions (Appendix A.2).
> - We added another experiment setting, the Ahmed body CFD dataset. On this high-resolution 3D task, we achieve 38.4% memory reduction while the error stays within 0.05% of the full-precision model.
>
> Thank you very much, once again, for your positive view of our work and your excellent suggestions. We would be happy to answer any additional questions.
> (2/2)

---

> > ### Comment · Reviewer_XXPr · 2023-12-04
> > **Read the author response**
> >
> > I have read the author response and would like to maintain my score.

---

### Author Response · Authors · 2023-11-17
**Author Response**

We thank all the reviewers for their insightful feedback and suggestions that helped significantly improve our mansucript. Our work theoretically and empirically demonstrates the power of a new mixed-precision training pipeline for neural operators. We appreciate that the reviewers find that our work is **novel** (XXPr, 5SPZ), gives strong **theoretical foundations** (XXPr, 5SPZ), gives thorough/impressive **experimental results** (XXPr, dRFB), and the application is of **significance** (XXPr, NM9S, 5SPZ). Furthermore, we are glad that the reviewers appreciated the clarity of our paper (dRFB, NM9S, 5SPZ) and that we open-sourced code (XXPr, NM9S).

We have now updated our paper to include all of the reviewers’ suggestions. We highlight the changes below
- Improved theoretical bounds: while we originally proved bounds using the Fourier basis, we have now proven matching results for arbitrary basis functions (Appendix A.2). This extends our results beyond Fourier neural operators to a larger class of integral operators, that can include e.g. Wavelet bases or graph neural operators.
- We added another experiment setting, the Ahmed body CFD dataset (Appendix B.1). On this high-resolution 3D task, we achieved 38.4% memory reduction while the error stays within 0.05% of the full-precision model (Figure 1 updated accordingly). Ahmed body CFD is the most memory-consuming application in our evaluation, costing 20,000 MB of GPU memory with batch size = 1.
- From additional experiments, we give even more evidence of how standard numerical stabilization techniques in mixed-precision training for neural nets, do not work for FNO training (Appendix B.5 and B.6).
- We give new experiments that directly compare our theoretical resolution and precision upper and lower bounds, to the empirical discretization and precision error (Appendix A.2). We find that the errors behave as expected by our theory, which is additional confirmation of the correctness of our theory and the applicability of our results.

Thank you once again, for your reviews, and we would be happy to answer any follow-up questions.

---

> ### Author Response · Authors · 2023-11-23
> **Update from authors**
>
> Dear Reviewers,
> Thank you once again for your suggestions. In addition to the above updates, we have also just added one more section (Appendix B.12) that presents an even more detailed ablation study for our method. This ablation study shows the benefit of (1) decomposition of einsum into sub-equations, (2) einsum path caching, (3) our greedy tensor contraction procedure, and (4) both weights and inputs in half-precision.
>
> Please let us know if you have any questions or follow-up comments to our responses, and we would be very happy to reply before the deadline tonight; thanks again!

---

### Meta-Review · Area_Chair_cir6 · 2023-12-11

**Metareview:**

The paper considers the effect of mixed-precision training of Fourier-based Neural Operators (FNO). The paper contributes both theoretical and empirical results. Theoretically, they show that the error from approximating the Fourier transform by a discrete Fourier transform is expected to be (in the worst case) of a much larger order (~1/eps^d) than the finite precision (~1/eps) under various Lipschitzness constraints. (Note, this is comparing an upper bound to an upper bound ---  there could be instances where either of these bounds is loose.)
Experimentally, the authors explore several (families of) PDEs, and show that a simple strategy of roughly performing the entire FNO block in "half-precision" (forward FFT, the tensor contraction, and the inverse FFT all in half precision) can substantially improve the speed and throughput of FNO (up to 1.5 times), substantially decrease memory (up to ~0.5) while sacrificing very little accuracy (~0.1%). Overall, I think the paper is well written, the experiments are thorough and convincing, and the question of understanding the effect of low-precision calculation in scientific AI applications is very timely and understudied.

**Justification For Why Not Higher Score:**

The paper has interesting results though the community of interest is fairly narrow (researchers interested in AI for science, and/or researchers working on understanding of numerics when training ML-based solvers for PDEs / AI for science more broadly).

**Justification For Why Not Lower Score:**

The paper has sufficiently many points of interest (error analysis of discretization and discrete fourier tranform; empirical analysis of the effect of half precision) to meet the ICLR bar.

---

### Decision · Program_Chairs · 2024-01-16

Accept (poster)